# The methyl-CpG-binding protein 2 inhibits cGAS-associated signaling

Hanane Chamma[1], Soumyabrata Guha[1], Roger Junior Eloiflin[1], Adeline Augereau [1], Pierre Le Hars[1], Moritz Schüssler [1], Yasmine Messaoud-Nacer[1], Mohammad Salma [1], Joe McKellar[1], Joanna Re[1], Morgane Chemarin[1], Arnaud Carrier[1], Michael A. Disyak[1], Clara Taffoni [1], Robin Charpentier[1], Zoé Husson[2], Emmanuel Valjent[3], Charlotte Andrieu-Soler[1,4], Eric Soler [1,4], Maria H. Christensen [5,6], Søren R. Paludan [6,7], Florian I. Schmidt [5], Daniela Tropea[8,9,10], Karim Majzoub [1], Isabelle K. Vila[1] & Nadine Laguette [1] ✉

The detection of cytosolic dsDNA by the cyclic GMP-AMP synthase (cGAS) is tightly regulated to avoid pathological inflammatory responses. Here, we show that the methyl-CpG-binding protein 2 (MeCP2), a major transcriptional regulator, controls dsDNA-associated inflammatory responses. The presence of cytosolic dsDNA promotes MeCP2 export from the nucleus to the cytosol where it interacts with dsDNA, dampening detection by cGAS. MeCP2 export partially phenocopies MeCP2 deficiency, leading to innate immune activation and enforcing an antiviral state. Finally, MeCP2 displacement from the nucleus following dsDNA stimulation disrupts its canonical function, leading to the reactivation of otherwise repressed genes, such as endogenous retroelements. Re-expression of the latter leads to the accumulation of DNA species feeding cGAS-dependent signalling. We thus establish a direct role of MeCP2 in the regulation of the breadth and nature of dsDNA-associated inflammatory responses and suggest targeting dsDNA-associated pathways or endogenous retroelements as therapeutic options for patients with MeCP2 deficiency.

Cytosolic dsDNAs are recognized drivers of chronic inflammation in a broad range of human diseases[1]. A major pathway involved in detecting cytosolic dsDNA and triggering the subsequent activation of inflammatory responses, relies on the cyclic GMP-AMP (cGAMP) synthase (cGAS) pathogen recognition receptor[2–5]. The interaction of cytosolic dsDNAs with cGAS leads to the production of the 2′3′-cGAMP second messenger, which in turn interacts with the stimulator of interferon genes (STING) adaptor protein, promoting the recruitment and activation of transcription factors, such as the Interferon regulatory factor 3 (IRF3), that drive a transcriptional programme ultimately leading to the production of type I Interferons (IFN) and inflammatory cytokines[2–5]. Regulation of the cGAS-STING signalling axis at several levels has been reported, including STING degradation[6], or regulators of cGAS catalytic activity[7,8]. In contrast, although the detection of dsDNA by cGAS is a key rate-limiting step in the activation of cGAS-STING signalling, few regulators of this interaction have been

[1]Institut de Génétique Moléculaire de Montpellier (IGMM), Université de Montpellier, Montpellier, France. [2]Institut de Génomique Fonctionnelle (IGF), Université de Montpelier, Montpellier, France. [3]Institut des Neurosciences de Montpellier (INM), Université de Montpellier, Montpellier, France. [4]Université de Paris, Laboratory of Excellence GR-Ex, Paris, France. [5]Institute of Innate Immunity, Medical Faculty, University of Bonn, Bonn, Germany. [6]Department of Biomedicine, Aarhus University, Aarhus, Denmark. [7]Center for Immunology of Viral Infections, Aarhus University, Aarhus, Denmark. [8]Trinity Translational Medicine Institute, St James Hospital, Dublin, Ireland. [9]FutureNeuro Research Ireland Centre, Trinity College Dublin, Dublin, Ireland. [10]Trinity College Institute for Neuroscience, Lloyd Building, Dublin, Ireland. ✉e-mail: nadine.laguette@cnrs.fr

identified to date[9]. Those include the Three-prime exonuclease 1 (TREX1), the ablation of which feeds cGAS-STING-dependent signalling through endogenous cytosolic DNA accumulation, and underlying the most severe form of Aicardi-Gouttière Syndrome (AGS)[10]. This highlights the importance of the regulation of the cytosolic DNA-cGAS interaction for the regulation of pro-inflammatory signalling.

Pathological activation of cGAS-STING signalling has been documented in a wide range of neurological disorders[11], mostly highlighting a link between chronic STING signalling and neurodegeneration[12–14]. Similarly, chronic inflammatory responses and immunological dysfunction have been reported in the Rett syndrome neurodevelopmental disorder (RTT; OMIM 312750). RTT is a rare genetic disorder frequently associated with de novo mutations in the *Methyl-CpG-binding protein 2* (*MECP2*) X-linked gene, giving rise to dysfunctional MeCP2[15]. Neurological manifestations of RTT have been shown to be reversible[16] and accompanied by immunological dysfunction as well as chronic low-grade inflammation[17–19]. Indeed, the absence of MeCP2 sensitizes immune cells to pro-inflammatory stimulation[20]. While inflammation has been proposed to be associated with RTT disease progression and severity[21,22], there is no mechanism explaining the onset of chronic inflammation in RTT.

MeCP2 is a broadly expressed nuclear protein known to operate as a major transcription regulator[23]. MeCP2 was initially described to bind and repress the expression of methylated regions of the genome[23,24], but the functions of MeCP2 in the regulation of the mammalian genome have expanded in recent years. Indeed, MeCP2 is now recognized to play context-dependent roles in gene expression and chromatin architecture regulation through DNA methylation-independent mechanisms[25–27]. MeCP2 was shown to associate with and repress genomic locations corresponding to endogenous retroelements, such as the Long Interspersed Nuclear Element-1 (LINE-1)[28]. Interestingly, increased LINE-1 activity has been shown to be sufficient to generate DNA substrates for cGAS-STING activation[29].

Altogether, the current state-of-the-art suggests a link between MeCP2 and dsDNA-mediated activation of the cGAS-STING pathway. Thus, we here investigated the potential role of MeCP2 as a regulator of dsDNA-associated inflammatory responses.

## Results

### MeCP2 interacts with cytosolic dsDNA

Previous work has shown that several actors of cytosolic dsDNA recognition, including cGAS, are primarily nuclear in the absence of cytosolic dsDNA[8,30], suggesting that dsDNA binders are capable of interacting with dsDNAs regardless of their subcellular localisation. We thus hypothesized that MeCP2 may also interact with cytosolic dsDNA. To test this hypothesis, we performed a series of DNA pulldown experiments.

First, whole cell extracts from wild-type (WT) mouse embryonic fibroblasts (MEF) were incubated with streptavidin beads alone, or streptavidin beads on which either 80nt-long 5'biotin-bearing ssDNA (b-ssDNA) or dsDNA (b-dsDNA) were immobilized (in vitro pulldowns), prior to assessment of bound proteins by Western blot (WB) (Fig. 1a). Both DNA probes are capable of inducing the expected cGAS-STING pathway activation, as characterized by increased cGas levels, Sting degradation, phosphorylating activation of Sting and Irf3 (Supplementary Fig. 1a) and increased expression of *Interferon β* (*Ifnβ*) and *Interleukin 6* (*Il6*), as well as IFN response genes, such as the *C-X-C motif chemokine ligand 10* (*Cxcl10*), *2′–5′-oligoadenylate synthetase 1* (*Oas1*) and *cGas* (Supplementary Fig. 1b). No significant modulation of *Mecp2* transcriptional levels were observed (Supplementary Fig. 1b). In vitro DNA pulldowns performed in WT-MEF using these probes showed robust recruitment of cGas and MeCP2 to dsDNA (Fig. 1b). Congruent with previous work, MeCP2 and cGas interaction with ssDNA was less efficient when compared to that with dsDNA (Fig. 1b)[31]. Thus, MeCP2 preferentially interacts with immune-stimulatory dsDNA.

We next wished to verify if the interaction of MeCP2 with dsDNA can be recapitulated in other cell types. To this aim, we performed in vitro DNA pulldowns using whole cell extracts from the WT-RAW264.7 murine myeloid cell line using bead-immobilized b-ssDNA and b-dsDNA. Both MeCP2 and cGas were recovered as dsDNA binders in this assay, while neither cGas nor MeCP2 was recovered as bound to ssDNA (Fig. 1c). To further assess the specificity of the binding of MeCP2 to dsDNA, we performed similar in vitro binding assays using bead-immobilized biotinylated dsRNA. We found that MeCP2 does not bind to dsRNA (Fig. 1d). Finally, we analysed whether the binding of MeCP2 to dsDNA may be sequence-specific by using another 80nt-long dsDNA synthetic probe. In vitro pulldown performed using whole cell extracts from WT-RAW264.7 showed a similar level of recruitment of MeCP2 and cGas to the 2 tested probes (Fig. 1e). Together, these experiments suggest that the ability of MeCP2 to interact with immune-stimulatory dsDNA is a conserved mechanism across cell types.

Next, to assess whether the recruitment of MeCP2 to dsDNA occurs in cells, WT-MEF were transfected or not with b-dsDNA for 6 hours, a time point at which immunofluorescence analyses showed that transfected dsDNA probes are prominently cytosolic (Supplementary Fig. 1c). Whole cell extracts were prepared and used in pulldowns using streptavidin affinity beads (in-cell pulldown, Fig. 1f). WB analysis of pulled-down material showed that MeCP2 and cGas interact with dsDNA in cells (Fig. 1g). Similar results were obtained in WT-RAW264.7 (Fig. 1h) and in the THP-1 human myeloid cell line (Supplementary Fig. 1d) when in-cell pulldowns were performed 6 hours after transfection of b-dsDNA. These data thus show that MeCP2 can interact dsDNA in MEF, as well as in immune murine and human cell lines. These data further suggest that the interaction takes place in the cytosol.

To assess where the interaction between MeCP2 and dsDNA probes takes place, we performed in vitro DNA pulldowns using cytosolic and nuclear extracts from WT-MEF (Supplementary Fig. 1e). WB analyses showed an enrichment of MeCP2 bound to dsDNA in both fractions (Supplementary Fig. 1f), supporting an interaction of cytosolic MeCP2 with dsDNA. In addition, we performed immunofluorescence assays where WT-MEF were challenged with dsDNA prior to staining using MeCP2 and dsDNA-specific antibodies. We observed that dsDNA challenge led to MeCP2-specific staining in the cytosol (Fig. 1i, j), with a significant overlap with the dsDNA signal (Fig. 1k). To assess whether MeCP2 cytosolic staining is specific to dsDNA as suggested by Fig. 1b–e, we next challenged WT-MEF with ssDNA or dsDNA prior to analysis of MeCP2 subcellular localization. Immunofluorescence analyses showed an increase of MeCP2 cytosolic levels, coupled to a decrease in MeCP2 nuclear levels upon challenge with dsDNA, and not with ssDNA (Fig. 1l, m). Transfection of dsRNA did not lead to an increase of MeCP2cytosolic staining (Supplementary Fig. 1g) while MeCP2 cytosolic staining was observed regardless of the dsDNA sequence used (Supplementary Fig. 1h). Finally, dsDNA transfection in WT-RAW264.7 also led to MeCP2 cytosolic staining (Supplementary Fig. 1i).

Altogether, these data support that the presence of MeCP2 in the cytosol is triggered by the presence of cytosolic dsDNA.

### MeCP2 is actively exported from the nucleus in the presence of cytosolic dsDNA

MeCP2 is mostly known for its nuclear localization and functions[32]. Thus, its presence in the cytosol is surprising. This led us to question whether dsDNA stimulation may trigger MeCP2 export from the nucleus.

To test whether increased cytosolic levels of MeCP2 result from active export from the nuclear compartment, we performed time course analyses of MeCP2 subcellular localisation following dsDNA stimulation for 3, 6, and 16 hours in WT-MEF. Immunofluorescence

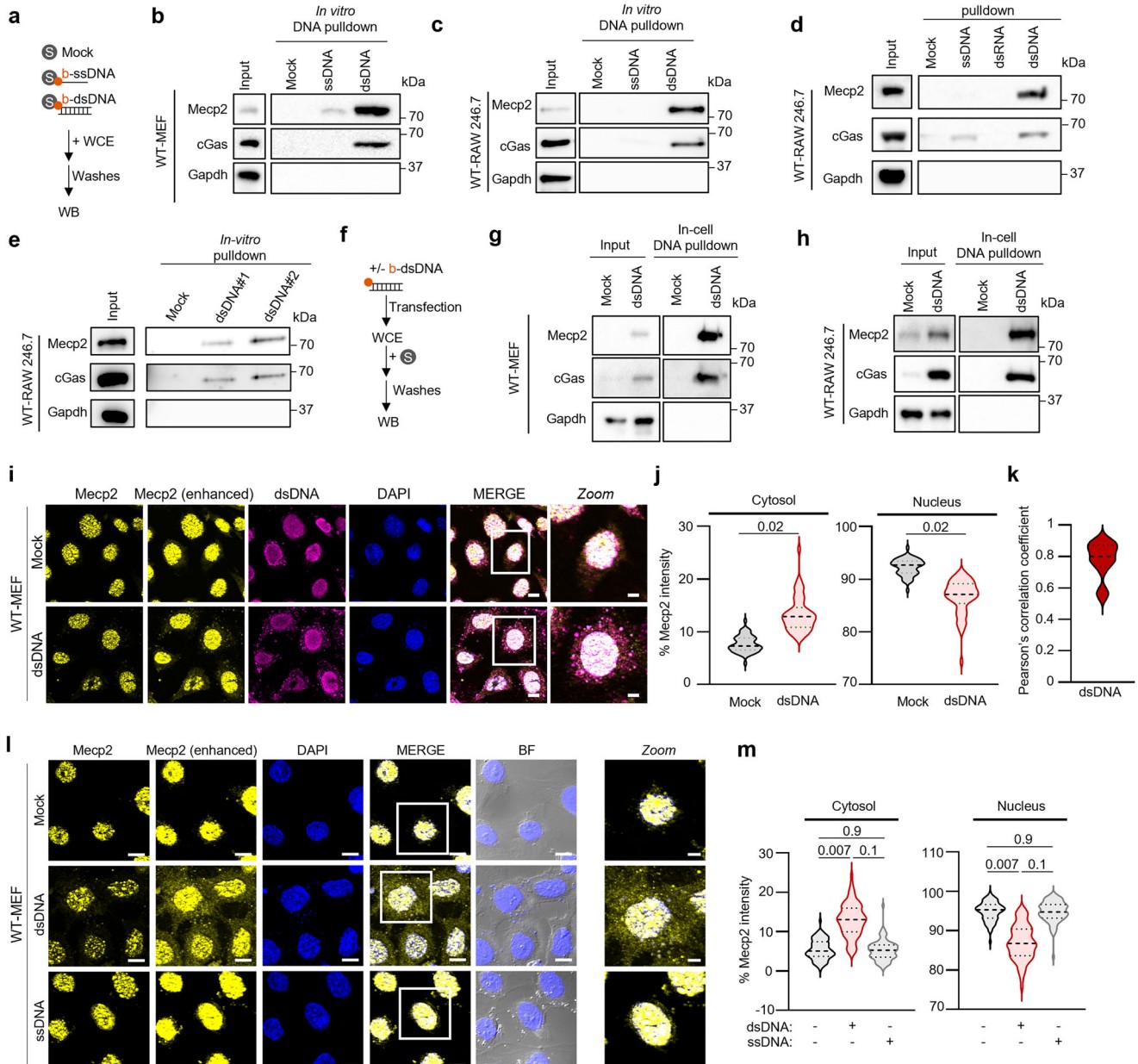

**Fig. 1 | MeCP2 interacts with cytosolic dsDNA. a** Experimental scheme for in vitro DNA pull-downs. **b** Whole cell extracts (WCE) from WT-MEF were incubated with streptavidin beads alone (mock) or with streptavidin beads-bound biotinylated dsDNA (b-dsDNA) or ssDNA (b-ssDNA) prior to in vitro DNA pull-down. Input and eluates were analyzed by Western blot (WB) using the indicated antibodies. **c** As in **a**, except that WCE from WT-RAW264.7 were used. **d** In vitro pulldown was performed as in **c**, except that beads-bound biotinylated dsRNA were also used. **e** In vitro pulldown was performed as in **b**, except that two different b-dsDNA were used. **f** Experimental scheme for in-cell DNA pulldowns. **g** WT-MEF were transfected with b-dsDNA or not (mock) before WCE preparation and streptavidin-affinity pull-down. Input and eluates were analyzed by WB using the indicated antibodies. **h** As in **g**, except that WCE were from WT-RAW264.7 cells transfected or not with b-dsDNA. **i** Immunofluorescence analysis was conducted on WT-MEF transfected or not with dsDNA for 6 h using anti-MeCP2 antibody, anti-dsDNA antibody and DAPI nuclear staining. BF, bright field. Images are representative of 3 independent experiments. Scale bar: 10 μm, except for Zoom: 5 μm. **j** Violin plots show the % MeCP2 intensity in the cytosol in experiments performed as in **i** (*n* = 123 for Mock and for 120 dsDNA-transfected cells). **k** Pearson's correlation coefficient was calculated on the cytosolic dsDNA and MeCP2 signals in WT-MEF treated as in **j**. **l** Immunofluorescence analysis of WT-MEF transfected or not with dsDNA and ssDNA for 6 h was performed using anti-MeCP2 antibody (enhanced signal or not) and DAPI nuclear staining. BF, bright field. Scale bars: 10 μm; Scale bars for Zoomed images: 5 μm. Images are representative of 3 independent experiments. **m** Violin plots show the % MeCP2 intensity in the cytosol and nucleus in experiments performed as in **l**; *n* = 105 cells per condition. WB and images are representative of at least 3 independent experiments. Significance was assessed using two-sided Student t-test. ns: non-significant. *$P < 0.05$, **$P < 0.01$, and ****$P < 0.0001$. Source data are provided as a Source Data file.

analyses showed that dsDNA stimulation led to increased cytosolic MeCP2 staining, accompanied by a decrease of MeCP2 nuclear staining (Fig. 2a, b). To control for the specificity of the cytosolic Mecp2 staining, similar experiments were performed in MEF expressing a MeCP2-targeting gRNA (MEF^gMecp2) (supplementary Fig. 2a, b), as well as in co-cultures of control gRNA expressing MEF and MEF^gMecp2 (supplementary Fig. 2c), showing that dsDNA stimulation induced MeCP2 cytosolic foci formation only in MeCP2 proficient cells.

Next, we performed subcellular fractionation experiments where the cytosolic and nuclear soluble fractions were isolated following

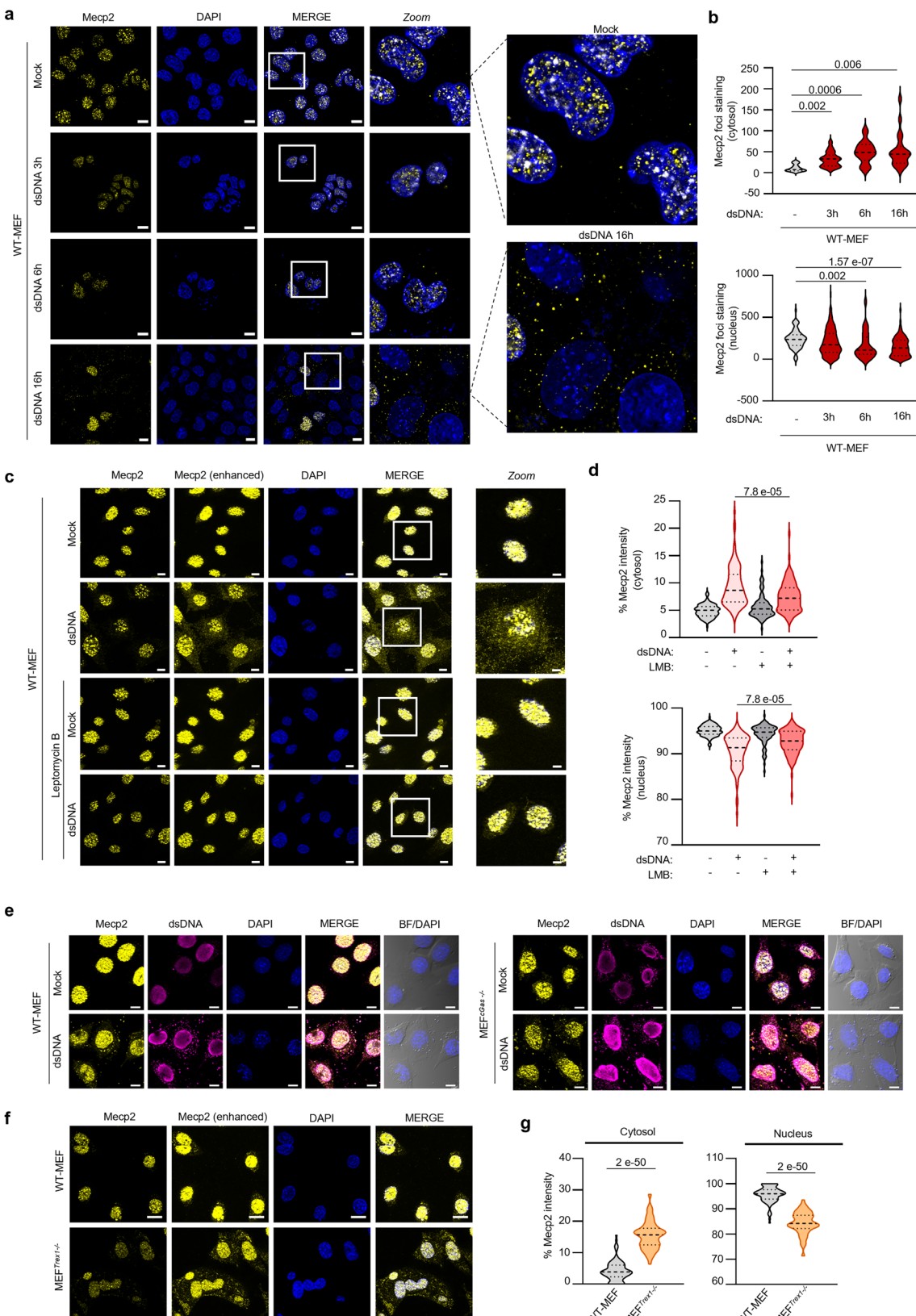

challenge with dsDNA. WB analyses showed an enrichment of MeCP2 in cytosolic fractions of WT-MEFs (Supplementary Fig. 2d). Similar subcellular fractionation was conducted in WT-RAW264.7, showing that dsDNA challenge induces increased MeCP2 signal in the cytosolic fraction (Supplementary Fig. 2e). Next, we performed immuno-fluorescence analyses of MeCP2 subcellular localization in WT-MEF

upon treatment with the Leptomycin B nuclear export inhibitor prior to stimulation or not with dsDNA. Leptomycin B treatment led to an efficient block of nuclear-cytosolic transport, as shown by immuno-fluorescence of RAN Binding Protein 1 (Ranbp1) (Supplementary Fig. 2f). As expected, Leptomycin B treatment also abolished dsDNA-induced cGas export (Supplementary Fig. 2g)[30]. We found that Leptomycin B

**Fig. 2 | Challenge with dsDNA triggers MeCP2 export. a** WT-MEF were stimulated with dsDNA for 3, 6, and 16 h prior to immunofluorescence analyses using an MeCP2 targeting antibody and DAPI nuclear staining. Scale bars: 10 μm. **b** Violin plots show the number of MeCP2 foci intensity in the cytosol and in the nucleus in experiments performed as in **a**; $n > 50$ cells per condition. **c** Immunofluorescence analysis was performed on WT-MEF treated or not with 20 nM of Leptomycin B (LMB) prior to dsDNA transfection for 3 h using anti-MeCP2 antibody (enhanced signal or not) and DAPI nuclear staining. Scale bars: 10 μm; Scale bar for Zoomed images: 5 μm. Images are representative of two independent experiments. **d** Violin plots show the % of MeCP2 intensity in the cytosol and nucleus in cells treated as in

**c**; $n = 95$ cells per condition. **e** Immunofluorescence analysis was conducted on WT-MEF and MEF$^{cGas-/-}$ cells transfected or not with dsDNA for 6 h using anti-MeCP2 antibody, anti-dsDNA antibody and DAPI nuclear staining. BF, bright field. Images are representative of 3 independent experiments. Scale bar: 20 μm.
**f** Immunofluorescence analysis was performed on WT-MEF and MEF$^{Trex1-/-}$ using anti-MeCP2 antibody and DAPI nuclear staining. Scale bars: 20 μm. Images are representative of two independent experiments. **g** Violin plots show the % of MeCP2 intensity in the cytosol and nucleus of cells treated as in **f**, $n = 97$ cells per condition. Significance was assessed using a two-sided Student $t$-test. Source data are provided as a Source Data file.

treatment reduced the levels of MeCP2 in the cytosol following dsDNA stimulation (Fig. 2c, d). Conversely, Leptomycin B treatment led to increased retention of MeCP2 in the nucleus following dsDNA challenge (Fig. 2c, d). These data support that cytosolic MeCP2 accumulation upon dsDNA challenge is driven by active export of MeCP2.

That MeCP2 is not displaced in the cytosol following ssDNA or dsRNA stimulations (Fig. 1l and Supplementary Fig. 1g) suggests that inflammatory signalling does not drive MeCP2 export. To confirm this observation, we assessed MeCP2 subcellular localization in a context where dsDNA-associated inflammatory responses are abrogated. We thus challenged WT-MEF and cGas-knockout MEFs (MEF$^{cGas-/-}$) with dsDNA. Immunofluorescence analyses of MeCP2 subcellular localization in response to dsDNA stimulation showed that MeCP2 is exported to the cytosol regardless of cGas expression (Fig. 2e). These data confirm that MeCP2 export is triggered by the presence of dsDNA in the cytosol and is not a response to cGas-Sting signalling.

Finally, we questioned whether cytosolic accumulation of endogenous DNA may trigger MeCP2 export. To this aim, we analyzed MeCP2 subcellular localization in MEFs harbouring an invalidating mutation in the *Trex1* gene, leading to chronic accumulation of cytosolic DNAs[33], notably owing to aberrant activity of the long interspersed element-1 (LINE-1) endogenous retroelements. Immunofluorescence analyses of control and Trex1-deficient MEFs (MEF$^{Trex1-/-}$) showed the presence of MeCP2 in the cytosol (Fig. 2f, g). Furthermore, treatment with the Tenofovir reverse transcriptase inhibitor, known to inhibit LINE-1 activity[34], led to decreased cytosolic dsDNA, cGas (Supplementary Fig. 2h) and MeCP2 stainings (Supplementary Fig. 2i, j). Therefore, the presence of endogenous cytosolic dsDNA is sufficient to trigger MeCP2 export.

## MeCP2 and cGas interact with the same dsDNA moieties in the cytosol

Experiments performed in Fig.1 and Supplementary Fig. 1 show that MeCP2 and cGas are both capable of binding cytosolic dsDNAs. We thus questioned whether they may bind the same dsDNA moieties.

To this aim, MEFs engineered to stably express an enhanced green fluorescence protein (EGFP)-tagged cGas allele (MEF$^{EGFP-cGas}$) were transfected or not with dsDNA prior to immunofluorescence analyses of MeCP2 and EGFP-cGas co-localization using Pearson's correlation coefficient. This showed that dsDNA transfection increased the colocalization between cGas and MeCP2 in the cytosol (Fig. 3a, b), suggesting a tripartite interaction between dsDNA, cGas and MeCP2.

We next interrogated the dynamics of the interaction of MeCP2 and cGas with cytosolic dsDNA. To this aim, WT-MEFs were transfected with b-dsDNA for up to 6 hours prior to whole cell extraction and in-cell DNA pulldowns using streptavidin affinity beads. WB analyses showed MeCP2 and cGas interaction with dsDNA as early as 30 min post transfection (Fig. 3c). This suggests similar interaction dynamics of MeCP2 and cGas with dsDNA. This was further tested by performing MeCP2 immunoprecipitation in WT-MEFs, transfected or not with dsDNA prior to assessment of cGas co-immunoprecipitation. We found that dsDNA transfection led to co-immunoprecipitation of cGas with MeCP2, as compared to mock transfection (Fig. 3d). Next, RAW264.7 expressing a MeCP2-targeting gRNA (RAW264.7 $^{gMecp2}$) were

engineered to stably express a FLAG-tagged MeCP2allele (FLAG-MeCP2) (Supplementary Fig. 3a), prior to transfection for 1, 3, 6 or 16 hours with dsDNA and FLAG immunoprecipitation. WB analyses showed that the interaction between MeCP2and cGas increased over the time course (Fig. 3e). As a control, we monitored levels of Isg15, which increased in input material, but did not co-immunoprecipitate with FLAG-MeCP2 (Fig. 3e), supporting the specificity of cGas recruitment. Thus, when taken together, these data support that the presence of cytosolic dsDNA triggers the formation of MeCP2, cGas and dsDNA-containing complexes.

The recruitment of cGas and MeCP2 to the same dsDNA molecules in the cytosol raises the possibility that they may compete for interaction. To test this, we first used WT-MEF and MEF$^{cGas-/-}$ to perform in vitro DNA pulldowns. WB analyses showed that the absence of cGas enhanced the recruitment of MeCP2 to dsDNA (Fig. 3f and Supplementary Fig. 3b). Conversely, we assessed whether MeCP2 regulates cGas recruitment to dsDNA. To this aim, we used MEFs expressing control (MEF$^{gCTRL}$) or MeCP2-targeting gRNAs (MEF$^{gMecp2}$) to perform in vitro dsDNA pulldowns. We found that reduced levels of MeCP2led to increased cGas recruitment to dsDNA (Fig. 3g). Finally, we performed in vitro pulldowns using Sting knockout MEFs. We found that the absence of Sting did not modify MeCP2 nor cGas recruitment dsDNA (Supplementary Fig. 3c). Together, these data show that MeCP2 and cGas can regulate their respective recruitment to dsDNAs, further supporting that cGas and MeCP2 are recruited to the same dsDNA molecules.

## MeCP2 inhibits cGas-dependent Type I Interferon responses
Our data show that the absence of MeCP2 leads to an increase of cGas recruitment to cytosolic dsDNA (Fig. 3), suggesting that the absence of MeCP2 may promote enhanced dsDNA-induced cGas-dependent signalling. While it was previously reported that the absence of MeCP2 correlated with increased inflammatory responses[17-22], a role of MeCP2 in regulating cGas- or dsDNA-associated inflammatory responses was not reported.

To assess whether MeCP2 may regulate cGas activity, we challenged MEF$^{gCTRL}$ and MEF$^{gMecp2}$ (Supplementary Fig. 4a) with dsDNA prior to assessment of cGas-dependent pathway activation. Immunofluorescence analyses of cGas subcellular localization in these cell lines showed that dsDNA stimulation enhanced cGas levels in the cytosol in MEF$^{gMecp2}$ as compared to MEF$^{gCTRL}$ (Fig. 4a). Assessment of intracellular 2'3'-cGAMP levels in MEF$^{gCTRL}$ and MEF$^{gMecp2}$ cell lines showed that low levels of MeCP2 led to higher levels of dsDNA-associated 2'3'-cGAMP (Fig. 4b). Analyses of the expression of genes classically associated with cGas activation, such as *Ifnβ*, *Il6*, *Cxcl10* and *C-C motif chemokine ligand 5* (*Ccl5*) (Fig. 4c), and of inflammatory cytokine production (Fig. 4d, e) showed that MeCP2 knockout led to their increased levels. Similarly, RAW264.7$^{gMecp2}$ (Supplementary Fig. 4b) presented heightened *Ifnβ*, *Isg15*, *Cxcl10* and *cGas* upon dsDNA stimulation, as compared to control cells (RAW264.7$^{gCTRL}$) (Supplementary Fig. 4c), accompanied by upregulation of pTbk1 and pSting, attesting to activation of Sting, as well as increased pStat1 levels, that attests to the production of bioactive IFNs (Supplementary Fig. 4d). Importantly, the inflammatory responses witnessed upon dsDNA stimulation in cells

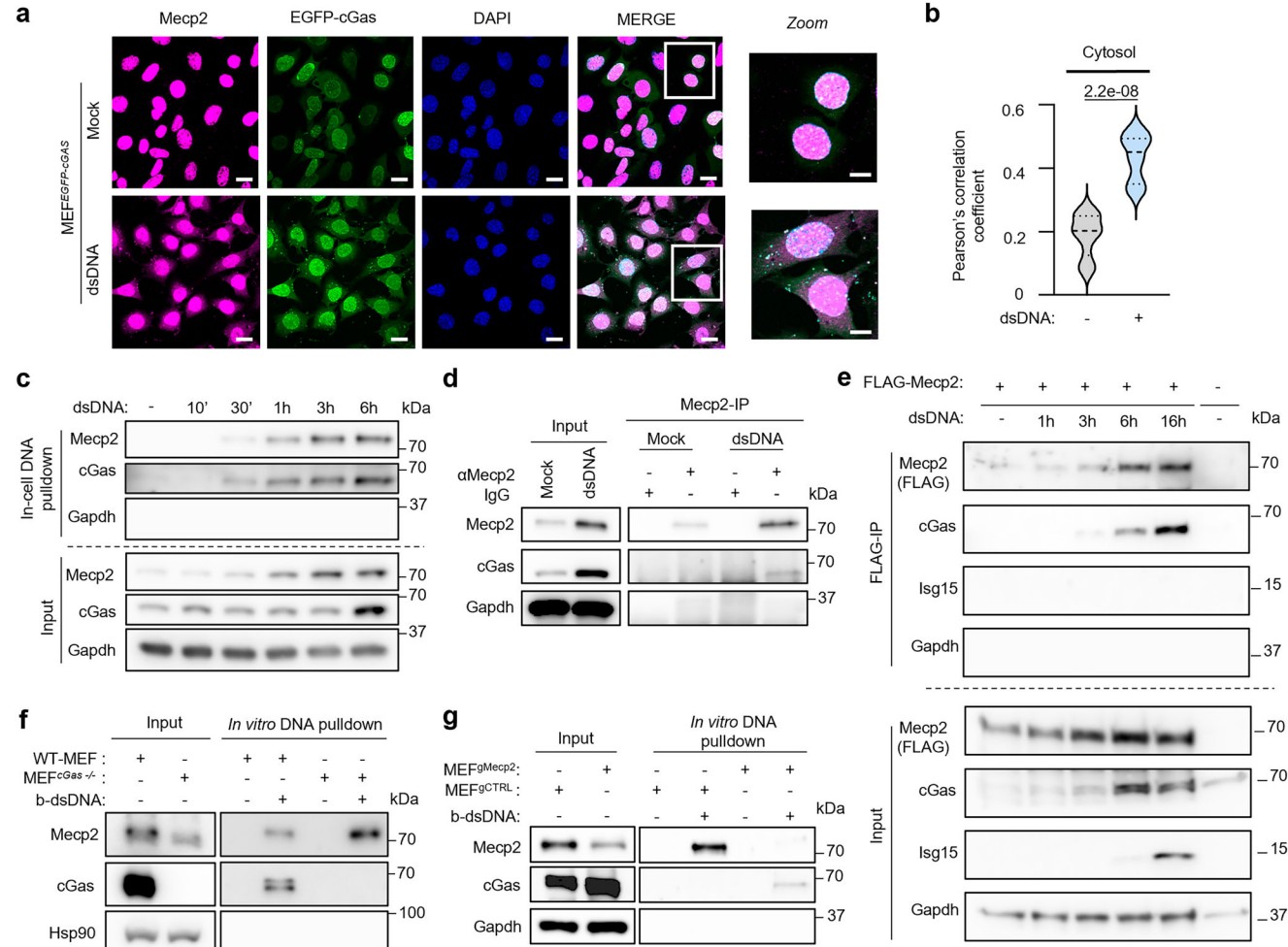

**Fig. 3 | Absence of MeCP2 increases cGas interaction with cytosolic dsDNA.**
**a** Immunofluorescence analysis was conducted on MEF stably expressing a EGFP-cGAS construct (MEF$^{EGFP\text{-}cGAS}$) transfected or not with dsDNA for 6 h using anti-MeCP2 antibody and DAPI nuclear staining. Images are representative of 3 independent experiments. Scale bar: 20 μm; Scale bar for Zoom: 10 μm. (Right) Graph presents mean ± SEM. **b** Pearson's correlation coefficient values for co-localization of cGas and MeCP2. *p* values were determined by Student's *t* test. ****$p < 0.0001$; $n = 12$. **c** WT-MEF were transfected or not for 10 min, 30 min, 1 h, 3 h or 6 h with b-dsDNA before whole-cell extract preparation and pull-down using streptavidin-affinity beads. Input and eluates were analyzed by WB using the indicated antibodies. **d** WT-MEF were transfected or not with dsDNA before whole-cell extract preparation and immunoprecipitation using control IgG or a MeCP2-specific antibody. Input and immunoprecipitated material were analyzed by WB using the

indicated antibodies. **e** MeCP2 knockout RAW264.7 were engineered to stably express FLAG-tagged MeCP2 prior to stimulation with dsDNA for 1, 3, 6 and 16 h. Whole cell extracts were subjected to FLAG immunoprecipitation prior to analysis of inputs and eluates by WB using the indicated antibodies. **f** Whole cell extracts prepared from WT-MEF or MEF$^{cGas\text{-}/\text{-}}$ cells were incubated with streptavidin beads alone or with streptavidin bead-bound b-dsDNA prior to pull-down. Input and eluates were analyzed by WB using the indicated antibodies. **g** Whole cell extracts prepared from MEF expressing a control non-targeting gRNA (MEF$^{gCTRL}$) or a MeCP2-targeting gRNA (MEF$^{gMecp2}$) were incubated with streptavidin beads alone or with streptavidin bead-bound b-dsDNA prior to pull-down. Input and eluates were analyzed by WB using the indicated antibodies. All WB are representative of 3 independent experiments. Source data are provided as a Source Data file.

expressing a MeCP2-targeting gRNA is decreased by treatment with the H-151 Sting inhibitor (Fig. 4f), suggesting that the witnessed inflammatory signature is, at least in part, Sting-dependent. Together, these data support the conservation of an inhibitory role of MeCP2 in different cell lines.

Next, we wished to confirm the direct role of MeCP2 in negatively regulating dsDNA-associated inflammatory responses. Using 2 different gRNAs targeting MeCP2 (Supplementary Fig. Se), we confirmed that absence of MeCP2 enhanced inflammatory gene expression (Supplementary Fig. 4f). Conversely, when MEFs were transfected with either control (Empty) or MeCP2-expressing vector (eMeCP2) (Supplementary Fig. S4g), we found that overexpression of MeCP2 prior to dsDNA stimulation led to decreased expression of *Ifnβ*, *Il6*, and *Cxcl10* as compared to Empty vector-expressing cells (Fig. 4g). We next questioned the specificity of the inhibitory role of MeCP2 and challenged MEF$^{gCTRL}$ and MEF$^{gMecp2}$ with the poly(I:C) synthetic RNA agonist

that stimulates cGas-independent inflammatory pathways. Absence of MeCP2 did not modulate the expression of Inflammatory genes following poly(I:C) stimulation (Fig. 4h). This confirms the role of MeCP2 as a negative regulator of cGas-associated inflammatory responses.

Taken together, these data (Figs. 3 and 4) show that the absence of MeCP2 leads to increased cGas recruitment to cytosolic dsDNA in cells and subsequent activation of cGas-dependent inflammatory responses.

## Absence of MeCP2 enforces an antiviral state
That MeCP2 inhibits dsDNA-associated inflammatory responses suggests that the absence of MeCP2 may lead to chronic low-grade type I IFN responses. The latter are known to lead to the establishment of protective antiviral states. This led us to question the interplay between viral infections and MeCP2 (Fig. 5a).

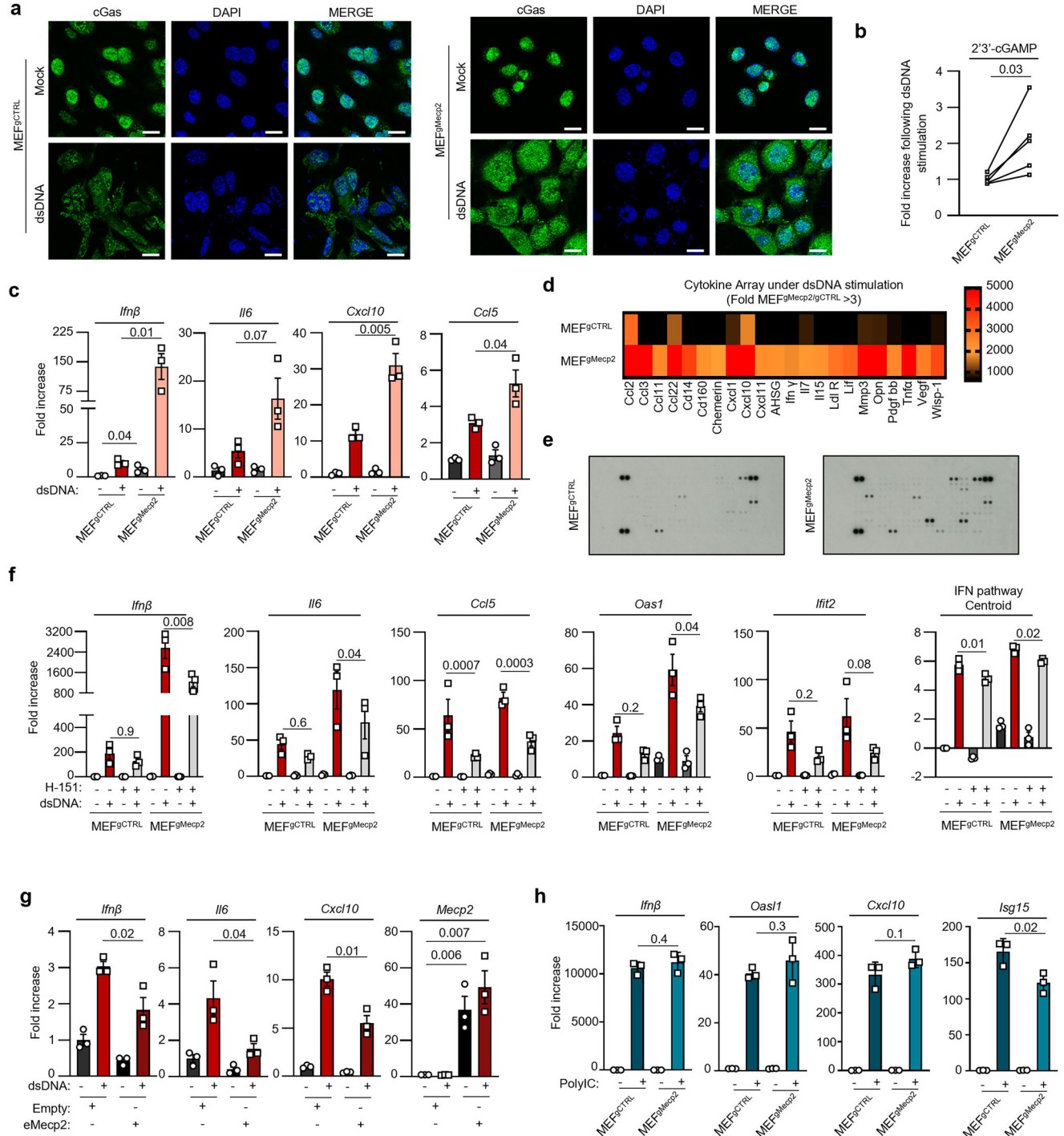

**Fig. 4 | Absence of MeCP2 primes cGas-dependent inflammatory responses.**
**a** Immunofluorescence analysis was conducted on MEF^gCTRL or MEF^gMecp2 after challenge or not with dsDNA for 6 h, using anti-cGas antibody and DAPI nuclear staining. Scale bar: 20 μm. Images are representative of 3 independent experiments. **b** Intracellular 2'3'-cGAMP levels were measured by ELISA following transfection or not with the dsDNA of MEF^gCTRL or MEF^gMecp2. Graph presents fold increase 2'3'-cGAMP levels from 5 independent experiments. **c** MEF^gCTRL or MEF^gMecp2 were challenged or not with dsDNA for 6 h prior to gene expression analyses. Graph presents mean (±SEM) *Ifnβ*, *Cxcl10*, *Il6* and *Isg15* mRNA levels (*n* = 3 independent experiments). **d** MEF^gCTRL or MEF^gMecp2 were challenged with dsDNA for 24 h prior to collection of the supernatant and analyses using proteome profiler antibody arrays. The heat map presents data obtained from duplicate measurements.

**e** representative images of arrays from **d**. **f** MEF^gCTRL or MEF^gMecp2 were challenged or not with dsDNA for 6 h in the presence or not of the H-151 Sting inhibitor. Graphs present mean (±SEM) *Ifnβ*, *Il6*, *Ccl5*, *Oas1*, and *Ifit2* mRNA levels and mean centroid analysis (*n* = 3 independent experiments). One-way ANOVA with Sidak's multiple comparison. **g** MEF overexpressing WT-MeCP2 (eMeCP2) or not (Empty) were transfected or not with dsDNA for 6 h prior to analysis of *Ifnβ*, *Il6*, *Cxcl10*, and *Mecp2* mRNA levels. Graphs present mean (±SEM) from 3 independent experiments.
**h** MEF^gCTRL or MEF^gMecp2 were challenged or not with poly(I:C) for 6 h prior to gene expression analyses. Graphs present mean (±SEM) *Ifnβ*, *Oasl1*, *Cxcl10*, and *Isg15* mRNA levels (*n* = 3 independent experiments). Significance was assessed using a two-sided Student *t*-test, except when otherwise stated. Source data are provided as a Source Data file.

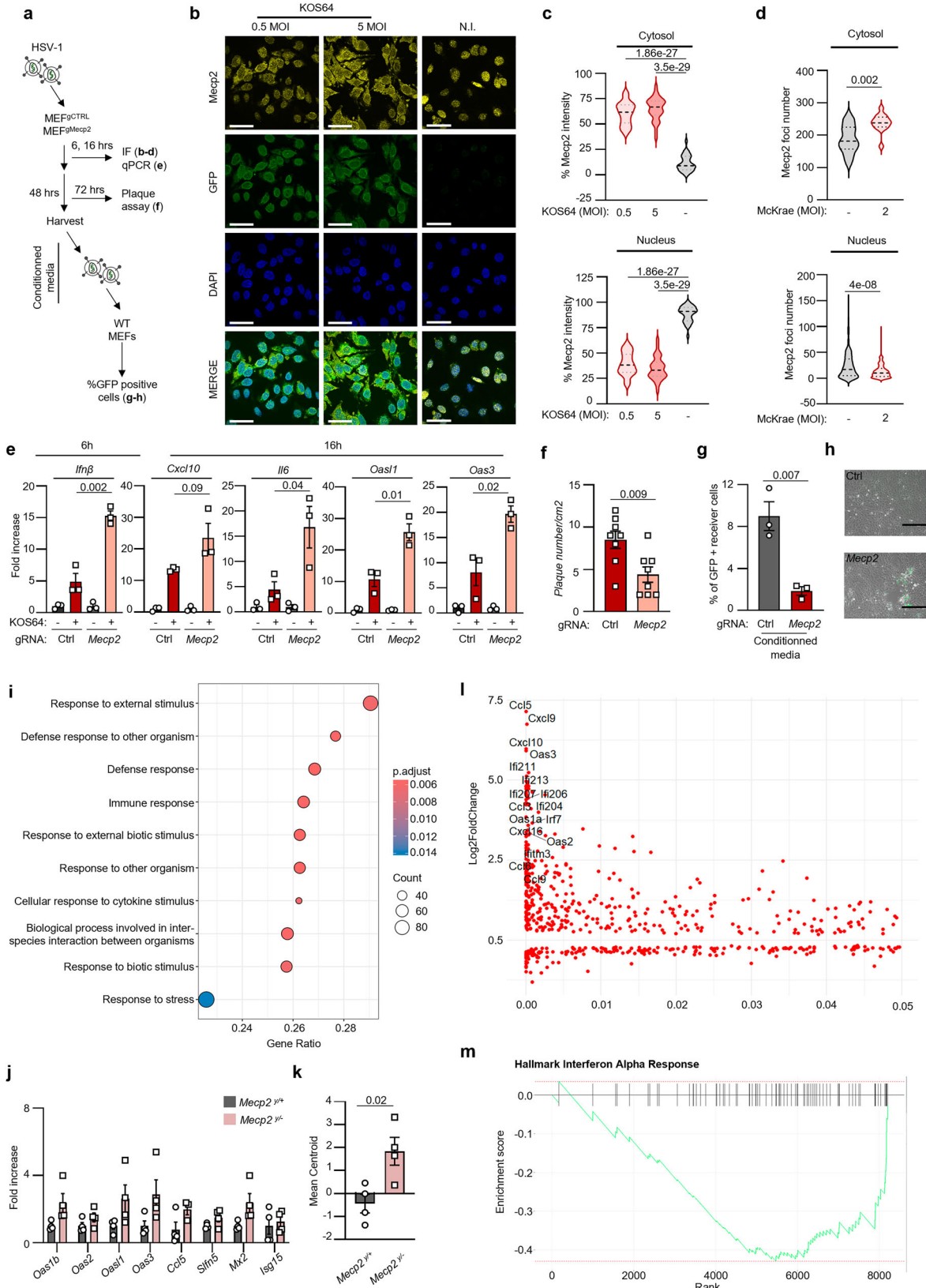

We first questioned whether the delivery of DNA in the cytosol through infection with a DNA virus would induce accumulation of MeCP2 in the cytosol, as witnessed upon dsDNA transfection (Fig. 2). We used two molecular clones of the Herpes simplex virus 1 (HSV-1), namely KOS64 and McKrae, harbouring dsDNA genomes. We found that infection of WT-MEF with HSV-1 led to the presence of MeCP2 in the cytosol (Fig. 5b-d and Supplementary Fig. 5a). Similar experiments were conducted using a molecular clone of the Vesicular stomatitis virus (VSV), a virus harbouring an ssRNA genome. In contrast to what was visualized following infection with HSV-1, we found that infection with VSV did not lead to cytosolic staining of MeCP2 (Supplementary Fig. 5b). These experiments, combined with data in Fig. 1, show that the

**Fig. 5 | Absence of MeCP2 enforces an antiviral state. a** Experimental scheme for **b**–**h**. **b** WT-MEF infected or not with a GFP-expressing HSV-1 KOS64 at 0.5 or 5 multiplicity of infection (MOI) were analyzed by immunofluorescence using anti-MeCP2 antibody and DAPI nuclear staining. Images are representative of 3 independent experiments. Scale bar: 50 μm. **c** Percent MeCP2 intensity in the cytosol in cells infected as in (**b**); *n* = 30 cells per condition. **d** As in **c**, except that cells were infected with the EGFP-expressing HSV-1 McKrae at 1 MOI; *n* > 50 cells per condition. **e** MEF$^{gCTRL}$ or MEF$^{gMecp2}$ were infected or not with 5 MOI of HSV-1-KOS64 for 6 and 16 h prior to analysis of expression of indicated genes. Graphs present the mean (±SEM) of 3 independent experiments. **f** Mean (±SEM) plaques per cm2 after 72 h of infection of MEF$^{gCTRL}$ or MEF$^{gMecp2}$ with 1 MOI of HSV-1 KOS64. 8 replicates, representative of 3 independent experiments. **g** WT-MEF were infected with conditioned media from HSV-1 KOS64-infected MEF$^{gCTRL}$ or MEF$^{gMecp2}$. The graph presents the mean (±SEM) percentage GFP-positive cells in recipient cells

(*n* = 3 independent experiments). **h** Representative images of cells in **g**. Scale bar: 400 μm. **i** Gene Set Enrichment Analysis (GSEA) was performed looking for Biological Process on DESeq2 results (log2foldchange > 0.01. GSEA *p*-value cut off = 0.05) from RNAseq data from RAW264.7$^{gCTRL}$ or RAW264.7$^{gMecp2}$. **j** Gene expression analysis was performed in the livers of male *Mecp2*$^{+/y}$ and *Mecp2*$^{-/y}$ mice. Graph presents mean (±SEM) fold increase gene expression in *Mecp2*$^{-/y}$ mice as compared to *Mecp2*$^{+/y}$; n = 4 mice per group. **k** Mean centroid expression was calculated on the expression of genes indicated in **j** (*n* = 4 mice per group). **l** Significant DEGs involved in inflammation, interferon-beta, virus response, STING, and innate immunity between hippocampi from control and *Mecp2*-silenced mice. X-axis: the False Discovery Rate (FDR). Gene symbols are reported for genes relevant to innate immunity. **m** Example Gene Set upregulated in patients with severe RTT symptoms. Significance was assessed using a two-sided Student T-test except for **h**, **j** and **k**. Source data are provided as a Source Data file.

delivery of dsDNA in the cytosol is necessary and sufficient to induce MeCP2 nuclear export.

Next, we assessed whether the presence or absence of MeCP2 can influence inflammatory responses following infection with HSV-1. To this aim, MEF$^{gCTRL}$ and MEF$^{gMecp2}$ were infected with HSV-1-KOS64 prior to assessment of the expression of *Ifnβ* at 6 hours and of antiviral IFN-stimulated genes such as *Cxcl10*, as 2′-5′-Oligoadenylate Synthetase-Like (*Oasl1*), 2′-5′-oligoadenylate synthetase 3 (*Oas3*) and *Il6* at 16 hours post infection. Absence of MeCP2 led to increased expression of those genes following HSV-1 infection (Fig. 5e). Thus, lower levels of MeCP2 lead to increased expression of inflammatory genes following infection with HSV-1, confirming the inhibitory impact of MeCP2 on cGas-associated signalling (Fig. 4).

We next assessed the capacity of HSV-1 to infect and replicate in MeCP2-deficient and proficient cells. To this aim, MEF$^{gCTRL}$ and MEF$^{gMecp2}$ were infected with HSV-1-KOS64 for 72 hours prior to assessment of plaque formation. Decreased levels of MeCP2 led to decreased plaque formation (Fig. 5f and Supplementary Fig. 5c), attesting to reduced ability of HSV-1 to infect these cells as compared to control cells. In contrast, when similar experiments were conducted using VSV, we found that the absence of MeCP2 did not significantly alter viral infection (Supplementary Fig. 5d). Next, conditioned media from HSV-1-KOS64-infected MEF$^{gCTRL}$ and MEF$^{gMecp2}$ were used to infect WT-MEF cells prior to quantification of infected cells. We found that infection of WT-MEF with the supernatant collected from MEF$^{gMecp2}$ led to a decreased number of infected cells as compared to supernatant collected from MEF$^{gCTRL}$ (Fig. 5g, h). Altogether, these data show that the absence of MeCP2 fosters an antiviral state that hinders infection by HSV-1. This further suggests that the absence of MeCP2 is sufficient to foster an antiviral state that is efficient towards DNA virus infection.

Since the presence of a type I IFN antiviral signature was not previously reported in MeCP2 deficiency, we finally assessed whether MeCP2 knockout is sufficient to promote the expression of a type I IFN signature. We first performed RNA sequencing (RNAseq) on RAW264.7$^{gCTRL}$ and RAW264.7$^{gMecp2}$. Gene Set Enrichment Analysis (GSEA) was conducted on genes upregulated in RAW264.7$^{gMecp2}$ as compared to RAW264.7$^{gCTRL}$, revealing an upregulation of processes related to stress response, immune response and biological interactions between organisms, including responses to external and biotic stimuli (Fig. 5i). We next assessed the presence of such an antiviral signature in the liver of mouse models of MeCP2 deficiency (*Mecp2*$^{-/-}$) as compared to WT littermates (*Mecp2*$^{y/+}$). Analyses of the expression levels of type I IFN response genes and antiviral genes (*Oas1b*, *Oas2*, *Oasl2*, *Oas3*, *Ccl5*, *Schlafen 5* (*Slfn5*), MX dynamin-like GTPase 2 (*Mx2*), and *Isg15* showed a tendency for increased expression (Fig. 5j). Mean centroid analysis of those antiviral genes showed a significant increase in MeCP2-deficient animals (Fig. 5k), confirming the presence of an antiviral type I IFN response in MeCP2 deficiency. Similarly, metanalyses were conducted on public datasets[35], generated from the hippocampus of adult mice where MeCP2 expression was disrupted by

RNA interference. Analyses focused on the Gene Ontology (GO) terms with the keywords "inflammation", "interferon-beta", "viral infection", "sting", and "innate immunity" in the description. We found that the differentially expressed genes (DEGs) comprise genes involved in the aforementioned groups (Fig. 5l and Supplementary Fig. 5e, f).

Finally, we wished to identify whether such an innate immune response signature could be observed in RTT patients presenting with MeCP2 deficiency. To this aim, we considered the samples taken from patients with RTT with different severity of symptoms[36]. GSEA showed enrichment of pathways related to response to IFN alpha (Fig. 5m), as well as response to IFNγ and STAT5 (Supplementary Fig. 5g, h), all hallmarks of inflammatory pathway activation and type I IFN signature. These findings are consistent with the inflammatory status detected in RTT patients[37].

Thus, our data show that the absence of MeCP2 leads to enhanced cGas activity, enforcing an antiviral state affecting dsDNA virus infection. These data further support that MeCP2 is a negative regulator of cGas activity.

## Displacement of MeCP2 from the nucleus disrupts its canonical function

We finally questioned whether MeCP2 nuclear export following dsDNA stimulation may disrupt its canonical gene expression regulation function. Owing to the abundance of Mecp2, the presence of multiple targets in the genome, and differential methylation patterns between cell types, identification of the precise regions regulated by MeCP2 has proven difficult. Notwithstanding, genes consensually identified as repressed by MeCP2 notably comprise genomic locations corresponding to endogenous retroelements[28,38,39]. We thus decided to assess LINE-1 expression following dsDNA stimulation to determine whether MeCP2's canonical function is altered.

To this aim, WT-RAW264.7 cells were transfected with dsDNA prior to RNAseq. The retrotranscriptome was analyzed using the Telescope software tool[28] in order to identify reads corresponding to endogenous retroelements. Principal component analysis was conducted on all expressed genes and on the retrotranscriptome, showing that mock and dsDNA-stimulated samples segregate into distinct subpopulations (Supplementary Fig. 6a, b). Volcano plots showed that, alongside the expected upregulation of gene expression following dsDNA stimulation (Supplementary Fig. S6c), dsDNA stimulation led to a robust upregulation of sequences corresponding to endogenous retroelements (726 unique transcripts; Fig. 6a). Obtained reads were normalized to their relative abundance in the genome prior to analysis of the global change in endogenous retroelement gene expression. This revealed a significant increase in reads corresponding to class I retroelements (Fig. 6b), which are those that replicate through RNA intermediates[29]. Significantly upregulated class I retroelements include long terminal repeats (LTR), endogenous retroelements (ERV) and non-LTR transposons. Non-LTR transposons significantly increased upon dsDNA stimulation comprise LINEs, but also Short interspersed

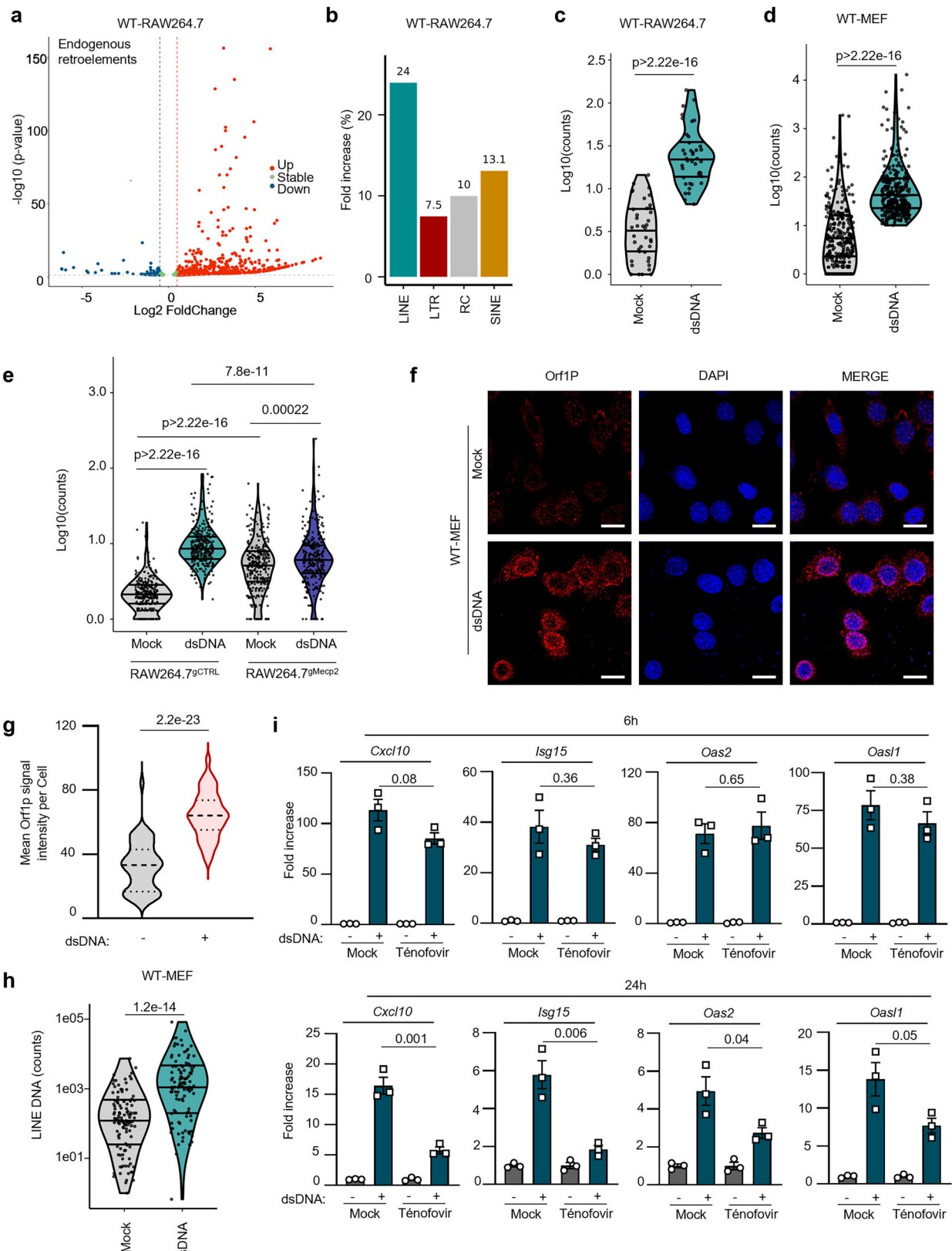

nuclear elements (SINEs) (Fig. 6b). We next examined more closely the upregulation of LINE-1 upon dsDNA stimulation and found that there is a significant increase of LINE-1 expression upon dsDNA stimulation as compared to the mock condition (Fig. 6c). Similar analyses were conducted on WT-MEF, showing that dsDNA transfection led to upregulation of transcripts corresponding to endogenous retroelements

(Supplementary Fig. 6d) with a marked increase of sequences corresponding to LINEs (Fig. 6d and Supplementary Fig. 6e).

We next questioned the contribution of MeCP2 displacement from the nuclear compartment to the observed increase of LINE-1 expression. To this aim, we performed RNAseq analyses following dsDNA stimulation in RAW264.7^gCTRL and RAW264.7^gMecp2. Similar to

**Fig. 6 | MeCP2 deficiency leads to the accumulation of immunogenic LINE-1-derived DNA. a** Volcano plot representing upregulated, stable or downregulated transcripts corresponding to endogenous retroelements WT-RAW264.7 stimulated with dsDNA for 6 h as compared to non-stimulated cells. **b** Graph presents the % fold increase of transcripts corresponding to type I endogenous retroelements in data from **a**. LINE: long interspersed nuclear elements, LTR: Long terminal repeats containing retroelements; ERV: Endogenous Retroviruses; SINE: short interspersed repetitive elements. **c** Violin plots present the transcripts corresponding to LINE-1 elements in data from **a**. **d** As in **c**, expect that WT-MEF were used for dsDNA stimulation. **e** As in **c**, except that RAW264.7[gCTRL] or RAW264.7[gMecp2] were used for dsDNA stimulation. **f** Immunofluorescence analyses were conducted on WT-MEF

transfected or not with dsDNA for 6 h using anti-Orf1p antibody and DAPI nuclear staining. Scale bar: 20 μm. Images are representative of 3 independent experiments. **g** Graph presents the mean Orf1p signal intensity quantified in images acquired as in **f**; $n = 49$ cells in the Mock condition and $n = 87$ in the dsDNA condition. **h** Cytosolic DNA isolated from WT-MEF stimulated or not with dsDNA for 6 h was sequenced using nanopore sequencing. The violin plot presents the reads corresponding to LINE-1. **i** WT-MEF were transfected or not with dsDNA for 6 or 24 h in the presence of the tenofovir reverse transcriptase inhibitor. The graphs present the mean (±SEM) fold increase of *Cxcl10*, *Isg15*, *Oas2*, and *Oasl1* as compared to untreated cells in 3 independent experiments. Significance was assessed using a two-sided Student *t*-test. ns: non-significant. Source data are provided as a Source Data file.

what was observed in WT-RAW264.7 (Fig. 6c), dsDNA stimulation led to increased LINE-1-associated transcripts (~3.1 folds, Fig. 6e). MeCP2-deficiency alone also led to an increase of LINE-1-associated transcripts levels (~2.3 folds), which were marginally increased following dsDNA stimulation (~1.1 folds, Fig. 6e). These data show that MeCP2 is involved in the regulation of LINE-1 expression upon dsDNA stimulation, supporting that MeCP2 displacement from the nucleus is involved in LINE-1 transcriptional activation upon dsDNA stimulation.

### LINE-1-derived dsDNA promotes cGAS-signalling

While MeCP2 has been reported to bind and repress LINEs, MeCP2 does not directly regulate SINEs[30]. SINEs are rather mobilised by LINE-1 activity[31]. The upregulation of SINEs (Fig. 6b) could therefore reflect the increased LINE-1 activity witnessed following dsDNA transfection, further suggesting that functional LINE-1-associated proteins may be produced following dsDNA stimulation. To test this hypothesis, immunofluorescence analyses were conducted to assess the levels of the Orf1p protein, which is an essential component of the LINE-1 retrotransposition machinery[32]. We found that stimulation with dsDNA was sufficient to lead to increased Orf1p levels in the cytosol of WT-MEFs (Fig. 6f, g) as well as in WT-RAW264.7 (Supplementary Fig. 6f, g). Infection with HSV-1 also led to increased cytosolic Orf1p staining (Supplementary Fig. 6h, i) in RAW264.7 cells (~2.3 fold), which is merely boosted in RAW264.7[Mecp2/-] (~1.2 fold). Therefore, our data indicate that dsDNA stimulation leads to increased expression of functional LINEs.

Interestingly, Orf1p bears reverse transcriptase activity. Therefore, increased Orf1p levels may lead to the accumulation of LINE-1-derived DNA in the cytosol. Such DNA species have been shown to be immune-stimulatory, raising the possibility that dsDNA stimulation may lead to an upregulation of LINE-1-derived DNA species that could sustain pro-inflammatory signalling. To test this hypothesis, we purified cytosolic DNA from WT-MEF stimulated or not with dsDNA and performed DNA sequencing using the nanopore technology. Analysis of LINE-1 sequences showed a 10-fold increase in the cytosolic fraction following dsDNA stimulation (Fig. 6h). Thus, dsDNA stimulation leads to de-repression of LINE-1 expression coupled to an increase of Orf1p and subsequent accumulation of LINE-1 DNA.

Finally, we tested the immunogenic potential of those DNA species by treating dsDNA-stimulated cells with the Tenofovir reverse transcriptase inhibitor. To this aim, WT-RAW264.7 cells were transfected or not with dsDNA in the presence or not of Tenofovir. Analyses of the expression of *Cxcl10*, *Isg15*, *Oas2*, and *Oasl1* were conducted at 6 and 24 hours post stimulation with dsDNA. This showed that, while Tenofovir treatment had a mild impact on the dsDNA-induced expression of those genes at 6 hours, 24 hours post-stimulation, treatment with Tenofovir dampened their induction (Fig. 6i). These data suggest that LINE-1-associated reverse transcriptase activity is involved in promoting persistent type I IFN responses. This further suggests that MeCP2 displacement from the nuclear compartment and the associated upregulation in LINE-1 activity serve to promote inflammatory signal persistence.

## Discussion

We here provide evidence for a role of MeCP2 as a negative regulator of dsDNA-induced inflammatory responses and show the presence of a type I IFN signature in in vivo models as well as in RTT patients samples. Indeed, we show that the presence of cytosolic dsDNA is sufficient to trigger MeCP2 re-localization to the cytosol, where it interacts with dsDNA and regulates cGAS-associated signalling, dampening type I IFN responses, but also de-repressing LINE-1 expression, thereby providing DNA substrates of cGAS-STING pathway activation.

This process modifies the cellular transcriptional landscape and primes the expression of genes that are otherwise repressed by MeCP2, yielding both the establishment of an antiviral state and the expression of transposable elements belonging to the abundant LINE-1 family[28,38,39]. LINE-1 transposons were previously shown to contribute to RTT onset and were suggested as causative of neurologic symptoms[40]. Interestingly, uncontrolled LINE-1 activity is known to promote the accumulation of cytosolic DNAs that trigger cGAS-STING-dependent inflammatory responses[29,33,41]. The de-repression of endogenous retroelements induced by dsDNA challenge that we measure in our experiments is both indicative of disruption of MeCP2 canonical function, but also serves to shape inflammatory responses, promoting their persistence. Furthermore, our data indicate that treatment with reverse transcriptase inhibitors that prevent LINE-1 DNA accumulation prevents the persistence of type I IFN responses. Knowing the deleterious impact of chronic type I IFN responses, this suggests that reverse transcriptase activity inhibition may permit resolution of the signalling and may be a treatment option to alleviate inflammation in RTT and therefore reduce the burden of RTT symptoms.

Interestingly, cGAS was shown to bind to pericentromeric regions of the genome[42], which are regions that comprise endogenous retroelement sequences[43]. That cGAS and MeCP2 can bind to the same dsDNA sequences in the cytosol thus raises the possibility that they may bind to overlapping regions of the genome. How this influences the detection of incoming viral dsDNA, such as in the case of HSV-1 infection, remains to be explored.

Additionally, our data reveal aberrant cytosolic MeCP2 subcellular localisation in the Trex1-deficiency model of type I interferonopathy. This suggests that chronic IFN responses witnessed in Trex1-deficiency may be a result of MeCP2-driven de-repression of LINE-1s, and subsequent activation of inflammatory and antiviral genes. In support of this, previous work has shown that treatment with reverse transcriptase inhibitors efficiently prevented endogenous retroelement activity in this model and decreased inflammatory responses and symptoms in vivo[34] and in patients[44]. Whether MeCP2 mislocalization plays a role in these contexts remains to be investigated. Furthermore, recent reports indicate that endogenous retroelement activity may be involved in the regulation of tonic inflammatory responses[45], suggesting that the absence of MeCP2 may contribute to priming inflammatory responses. Our study shows that MeCP2 ablation led to the upregulation of antiviral genes capable of limiting replication of the HSV-1 dsDNA virus. That replication of the VSV single-stranded RNA virus is not altered by the absence of MeCP2 also reinforces that MeCP2 ablation primes cells for dsDNA-associated type I IFN

responses. Alternatively, this may also suggest that the antiviral status elicited by the absence of MeCP2 can be overcome by RNA viruses. Indeed, our study does not rule out that RNA viruses may exploit MeCP2 during their life cycle. Furthermore, the current view suggests that ssRNA viruses efficiently overcome or circumvent cGas-Sting activation[46]. This further supports our evidence that cGas-Sting activity drives the antiviral status in MeCP2 deficiency.

Finally, we report the presence of an antiviral type I IFN signature in the liver of in vivo models of RTT. Interestingly, metabolic dysfunctions were previously reported in the livers of MeCP2-deficient mice[47]. In light of previous work showing that Sting is a crucial regulator of polyunsaturated fatty acid (PUFA) metabolism and that excess PUFA can inhibit Sting activity[48], it can be speculated that metabolic alterations observed in MeCP2-deficient mice livers may be attributed to chronic cGas-Sting activation driving PUFA metabolism imbalances. In this line, it has already been proposed that supplementation in Omega-3 polyunsaturated fatty acids may alleviate symptoms of RTT[22]. It is therefore tempting to propose that Sting pharmacological inhibition may bear promises for RTT patients.

## Methods

### MeCP2-deficient mouse models
*Post mortem* samples from MeCP2-deficient (Jackson (B6.129P2(c)-Mecp2tm1–1Bird) male mice between 8-12 weeks of age were a kind gift from Emmanuel Valjent (IGF Montpellier) and Adrian Bird (University of Edinburgh). Meta-analyzed mouse mRNA sequencing data[35] were from hippocampal mRNA profiles of 3-month-old mice following delivery of either a control shRNA sequence or a MeCP2-specific shRNA.

### Cells and cell culture
Wild-type (WT) murine embryonic fibroblast (MEF) and cGAS-deficient MEF (MEF$^{cGas-/-}$) were a gift of S. R. Paludan. Trex1-deficient MEF (MEF$^{Trex1-/-}$) were a gift from J. Rewhinkel. BHK21 cells were a gift of O. Moncorgé (IRIM, Montpellier). Parental RAW264.7 (*Mus musculus*), NIH3T3 (*Mus musculus*), HEK293T (*Homo sapiens*), and THP-1 (*Homo sapiens*) cells were obtained from the American Type Culture Collection (ATCC). MEF$^{gMecp2}$ and RAW264.7 $^{gMecp2}$ cell lines, along with their respective controls, were generated in the laboratory using the CRISPR-Cas9 technology. MEF, HEK293T, and RAW264.7 cells were maintained in Dulbecco's Modified Eagle Medium (DMEM) supplemented with 10 % Foetal Bovine Serum (FBS, Eurobio), 1% L-glutamine (Lonza), 1% Penicillin/Streptomycin (Lonza). In-house-generated knockout cell lines were maintained in the presence of the puromycin selection antibiotic. THP-1 cells were cultured in Roswell Park Memorial Institute (RPMI, Lonza) supplemented with 10% FBS, 1% penicillin/streptomycin and 1% L-glutamine. All cell lines were maintained at 37 °C, under 5% CO$_2$.

### Plasmids, constructs, and synthetic nucleic acid probes
MEF $^{gMecp2}$ and RAW264.7 $^{gMecp2}$ and control cell lines were generated by lentiviral transduction followed by selection, using the LentiCRISPRv2puro plasmid (Addgene #52961) in which control non-targeting or *Mecp2*-targeting guide RNAs (gRNA) were cloned using the following sequences:

### Control non-targeting gRNA
Forward (F): CACCGACGGAGGCTAAGCGTCGCAA;
Reverse (R): AAACTTGCGACGCTTAGCCTCCGTC

### *Mecp2*-targeting gRNA
F: CACCGCGCTCCATTATCCGTGACCG;
R: CGCGAGGTAATAGGCACTGGCCAAA

To generate the FLAG-tagged MeCP2 expressing construct, the MeCP2 gene was amplified by PCR from peGFP-N1-MeCP2 WT plasmid (Addgene #110186) using the Phusion High-Fidelity DNA polymerase kit (M0530L) followed by cloning into the pOZ vector[49].

Synthetic nucleic acid probes used for in vitro and in-cell pulldowns were purchased as ssDNA 80 base pair (bp)-long ssDNA probes, bearing or not 5' biotin or Cyanin 3-labels, from Integrated DNA Technologies (IDT). The 5' biotin and Cyanin 3 tags were always on the sense strand. When dsDNA probes were used, they were generated by annealing of sense and anti-sense ssDNA probes and integrity assessed as described in ref.[50]. In brief, sense and anti-sense ssDNA probes were annealed in a reaction combining equal amounts of the complementary ssDNA probes in annealing buffer (60 mM NaCl, 10 mM Tris-HCl, and 200 µM EDTA) using a temperature gradient from 95 °C to 4 °C. After annealing, the integrity of the dsDNA probes was assessed using 10% acrylamide gel electrophoresis in TBE (Tris-Borate-EDTA) buffer and visualization using ethidium bromide or SYBR safe under UV light.

Sequences of nucleic acid probes used in the present study are:
**Sense (S) ssDNA (S-ssDNA):** ACATCTAGTACATGTCTAGTC AGTATCTAGTGATTATC
TAGACATACATGATCTATGACATATATAGTGGATAAGTGTGG.
**Anti-sense (AS) ssDNA (AS-ssDNA):** CCACACTTATCCAC TATATATGTCATAGATCAT
GTATGTCTAGATAATCACTAGATACTGACTAGACATGTACTAG ATGT.

More precisely, dsDNA was obtained by annealing of S-ssDNA and AS-ssDNA; 5'biotin-bearing dsDNA (b-dsDNA) was obtained by annealing 5'-biotinylated S-ssDNA and AS-ssDNA; and Cy3-labelled dsDNA (Cy3-dsDNA) was obtained by annealing Cyanin3-labelled S-ssDNA and AS-ssDNA.

In the case where dsDNA1 and dsDNA#2 were used, dsDNA#1 corresponds to the annealing of the above-stated sequences, while the sequences annealed to obtain dsDNA#2 are:
S-ssDNA#2:
GACTGACTGACTGACTGACTGACTCCAGCCCGGCCCGACCCGACCG CACCCGGCGCGACTGACTGACTGACTGACTGACT
AS-ssDNA#2:
AGTCAGTCAGTCAGTCAGTCAGTCGCGCCGGGTGCGGTCGGGTCG GGCCGGGCTGGAGTCAGTCAGTCAGTCAGTCAGTC

### Generation of knock-out cell lines
Lentiviral particles used to generate MEF$^{gMecp2}$, RAW264.7$^{gMecp2}$, and control cell lines were produced by co-transfection of $2 \times 10^6$ HEK293T with 5 µg of LentiCRISPRv2puro plasmid expressing non-targeting or *Mecp2*-targeting gRNAs, together with 5 µg of psPAX2, and 1 µg of pMD2.G, using the calcium phosphate transfection protocol. Lentiviral particles were harvested 48 h after transfection and filtered (0.45 µM) prior to transduction of MEF or RAW264.7. Selection was initiated 72 h post-transduction using 1.5 µg/ml or 4 µg/ml puromycin for MEFs or RAW264.7, respectively. Protein levels of MeCP2 were controlled by Western blot (WB) after 3 days of selection.

### Transfection and treatments
For gene expression analyses, cells were transfected or not with 0.2 or 2 µg of ssDNA or dsDNA or 2 µg of poly(I:C) per well of 6-well plates. For in-cell pulldowns, cells were transfected or not with 1 or 10 µg of b-ssDNA, or b-dsDNA when experiments were conducted in 10 cm dishes. Two or 20 µg of nucleic acid probes were transfected when experiments were conducted in 15 cm dishes. Transfections were performed using JetPrime (Polyplus) in DMEM or Opti-MEM (GIBCO) for 1, 3, and/or 6 hours as indicated in the figure legends.

For nuclear export inhibition, cells were treated with 20 nM of Leptomycin B (LMB, Cell Signaling Technology (CST)) for 1 hour prior to dsDNA transfection for 3 hours. For STING inhibition, cells were treated with 1 µM of H-151 for 1 hour prior to dsDNA transfections for 6 hours. Reverse transcriptase activity was inhibited by treating cells

with 25 μM of Tenofovir for 24 hours prior to dsDNA transfection for 6 or 24 hours. Following transfection or treatment, cells were collected for gene expression analysis, WB, in-cell pulldowns, immunoprecipitations or cGAMP quantification, or fixed for immunofluorescence.

## RNA extraction and gene expression analyses

Total RNA was isolated with TRIzol reagent (Invitrogen) or using the GenElute™ Mammalian total RNA kit (Sigma #SLBW4972). RNA concentration was measured with a Nanodrop spectrophotometer (ND-1000, Nanodrop Technologies), prior to treatment with TURBO DNase (Ambion) and cDNA synthesis from 1-2 μg RNA using SuperScript IV (Invitrogen) using Oligo(dT) and quantification with a LightCycler 480 cycler (Roche) using SYBR Green Master Mix (Takara) and appropriate primers. Relative quantities of the transcripts were calculated using the ΔΔCt method, using the heat shock protein 90 (*Hsp90*) or glyceraldehyde-3-phosphate dehydrogenase (*Gapdh*) for normalization. The RT-qPCR primers used are listed below:

*Ccl5*: **F**: CAGCAAGTGCTCCAATCTTGC; **R**: CCACTTCTTCTCTGGG TTGGC

*Cxcl10*: **F**: ATGACGGGCCAGTGAGAATG; **R**: TCAACACGTGGGCA GGATAG

*Gapdh*: **F**: TTCACCACCATGGAGAAGGC; **R**: GGCATCGACTGTGGT CATGA

*Hsp90*: **F**: GTCCGCCGTGTGTTCATCAT; **R**: GCACTTCTTGACGAT GTTCTTGC

*Ifnβ*: **F**: CTGCGTTCCTGCTGTGCTTCTCCA; **R**: TTCTCCGTCATCT CCATAGGGATC

*Il6*: **F**: GACTTCCATCCAGTTGCCTTCT; **R**: TCCTCTCCGGACTTGT GAAGTA

*Isg15*: **F**: GTGCTCCAGGACGGTCTTAC; **R**: CTCGCTGCAGTTCTG TACCA

*Mecp2*: **F**: TCAGAAAGCTCAGGCTCTGC; **R**: CCCGGTCACGGATAA TGGAG

*Mx2:* **F**: CCTATTCACCAGGCTCCGAA; **R**: CGTCCACGGTACTG CTTTTC

*Oas1*: **F**: TGCATCAGGAGGTGGAGTTTG; **R**: ATAGATTCTGGGATC AGGCTTGC

*Oas1b*: **F**: GCAAAGGCACCACACTCAAG; **R**: CTCTCATGCTGAAC CTCGCA

*Oasl1*: **F**: CAGGAGCACTACAGACGTGG; **R**: GGTTACTGAGCCCAA GGTCC

*Oas2:* **F**: GAGTGGGAGGTGACGTTTGA; **R**: GAGTGGGAGGTGACG TTTGA

*Oas3:* **F**: CCAAAGCGTGGACTTTGACG; **R**: GCAGCTCTGTGAAGCA GGTA

*Slfn5:* **F**: CGAAATCATCTCGCAAGCCG; **R**: TGGTGGCAGATTCAAG CCAA

## Whole cell lysate preparation

For WB, pulldowns and immunoprecipitations, cells were harvested on ice in cold phosphate-buffered saline (PBS) using a cell scraper and lysed in 5 packed cell volume (PCV) of TENTG-150 [20 mM tris-HCl (pH 7.4), 0.5 mM EDTA, 150 mM NaCl, 10 mM KCl, 0.5% Triton X-100, 1.5 mM MgCl₂, and 10% glycerol], or with RIPA buffer [50 mM Tris-HCl (pH8.0), 150 mM NaCl, 1% NP-40, 0.5% sodium deoxycholate, 0.1% SDS], supplemented with 10 mM β-Mercaptoethanol, 0.5 mM phenylmethylsulfonyl fluoride (PMSF) and phosphatase inhibitor (PhosphoSTOP, Sigma-Aldrich) for 30 min at 4 °C. Cell lysates were centrifuged at 12,000 g for 30 min at 4 °C and supernatants were collected. Protein concentration was determined using the Bradford assay (Sigma-Aldrich).

## Nuclear-cytoplasmic fractionation

For subcellular fractionation, cells were harvested and pellet size normalized prior to lysis in 5 PCV of low salt buffer [100 mM NaCl, 0.1% Triton, 20 mM Tris pH 7.4, 2 mM MgCl2, 0.5 mM EDTA, and 10%

Glycerol], extemporaneously supplemented with 0.2 mM PMSF and 10 mM β-Mercaptoethanol, for 20 min on wheel at 4 °C. Nuclei were pelleted at 2,000 g for 10 min and cytosolic extracts were collected in a fresh tube. Nuclei were then lysed by adding 5 PCV high salt buffer [340 mM NaCl, 0.1% Triton, 20 mM Tris pH 7.4, 2 mM MgCl₂, 0.5 mM EDTA, and 10% Glycerol], extemporaneously supplemented with 0.2 mM PMSF and 10 mM β-Mercaptoethanol for 30 min on a wheel at 4 °C. Soluble nuclear extracts were collected after centrifuging the samples at 12,000 g for 10 min. When the Dignam nuclear and S100 extracts were used, they were prepared using a protocol adapted from ref. [49]. In brief, cells were washed with PBS and detached with PBS containing 1 mM EDTA for 5 minutes at room temperature. After centrifugation at 750 rpm for 1 minute at 4 °C, pellets were pooled and resuspended in 2X volume of hypotonic buffer (10 mM Tris pH 7.4, 1.5 mM MgCl2, 7.5 mM KCL, supplemented with 10 mM β-mercaptoethanol and 0.5 mM pMSF). Cells were incubated on ice for 10 minutes, and homogenized using a 15 ml glass Dounce homogenizer (pestle B). Lysates were centrifuged at 2500 rpm for 15 minutes at 4 °C, and the supernatant (cytoplasmic fraction) was collected while avoiding the lipid layer, and supplemented with 0.1 volume of 10X buffer (300 mM Tris pH 7.4, 30 mM MgCl2, 1.4 M KCL, freshly supplemented with β-mercaptoethanol and 0.5 mM pMSF). Ultracentrifugation was performed at 13,000 rpm for 2 hours at 4 °C, and the clean cytosolic supernatant was collected, snap-frozen in liquid nitrogen, and stored at -80 °C. Remaining pelleted nuclei were resuspended in 4X volume of high salt buffer 2 (20 mM Tris pH 7.4, 25% glycerol, 1.5 mM MgCl2, 0.2 mM EDTA, 0.340 M KCL, supplemented with 10 mM β-mercaptoethanol and 0.5 mM pMSF) and incubated for 30 minutes at 4 °C with gentle rotation. Following incubation, samples were centrifuged at 13,000 rpm for 30 minutes at 4 °C. The resulting nuclear fraction was collected, snap-frozen in liquid nitrogen, and stored at −80 °C. For dialysis, frozen cytoplasmic and nuclear extracts were thawed, centrifuged at 13,000 rpm for 30 minutes at 4 °C, added to the dialysis tubing, and dialyzed against 20X volume of dialysis buffer (20 mM Tris pH 7.4, 20% glycerol, 0.2 mM EDTA, 100 mM KCL, supplemented with 10 mM β-mercaptoethanol and 0.5 mM PMSF) for 4 hours at 4 °C under constant agitation. The dialysate was clarified by centrifugation, aliquoted, snap-frozen, and stored at −80 °C. 0.1% Triton X-100 was added to the lysates prior to use.

## Western Blot analyses

Protein samples were prepared in Laemmli buffer and heated at 95 °C for 5 min prior to resolution by sodium dodecyl sulfate–polyacrylamide gel electrophoresis (SDS-PAGE) using precast 10% or 12% gels (Invitrogen Novex Tris-glycine) followed by transfer onto nitrocellulose membranes using the Trans-Blot Turbo Transfer System (Biorad). Proteins were visualized on membranes using Ponceau S solution (Sigma-Aldrich) prior to 30 min blocking with PBS containing 0.1% Tween (PBS-T) supplemented with 5% milk, and incubation overnight at 4 °C for phosphorylated proteins or 90 min at room temperature with primary antibodies in 5% milk/PBS-T or 5%BSA/ PBS-T. Primary antibodies used include: anti-MeCP2 (1:500; Cell signalling D4F3, # 3456T), anti-cGas (1:1000; Cell signalling D3080, # 31659S), anti-phosphorylated Irf3 (1:500; Cell Signalling 4D4G, # 4947S), anti-Irf3 (1:1000; Cell Signalling D83B9,#4302), anti-phosphorylated Tbk1 (1:1000; Cell Signalling D52C2, # 5483S), anti-Tbk1 (1:1000; Cell Signalling D1B4, 3 3504S), anti-Sting (1:1000; Cell Signalling D2P2F, # 13647S), anti-phosphorylated Sting (1:1000; Cell Signalling D8F4W, # S365), anti-Hsp90 (1:1000; Cell Signalling C45G5, #4877), anti-Gapdh (1:5000; Proteintech Europe # 60004-1-Ig), anti-Lamin B1 (1:1000, Santa Cruz Biotechnology # sc-374015), anti-Acetylated Histone H3 (1:1000; Santa Cruz Biotechnology # sc-56616), anti-Flag (1:1000; Sigma # F1804), anti-Tubulin α (# 66031-1-Ig, Proteintech Europe, 1:10,000), and anti-Ranbp1 (1:100; Santa Cruz Biotechnology # sc-374352). Membranes were incubated with

Horseradish peroxidase (HRP)-coupled secondary antibodies (Cell Signalling, anti-rabbit # 7074, anti-mouse # 7076) at 1:2000 dilution for 1 hour at room temperature. Immunoreactivity was detected by Chemiluminescence (SuperSignal West Pico or Femto Thermo Scientific). Images were acquired on a ChemiDoc (Bio-Rad) or Amersham bioluminescence detection imager. Source data for Western blots are provided in the Source Data file and in the Supplementary Information file.

### In vitro biotinylated nucleic acid pull-down

In vitro pull-downs were performed using either whole cell lysates or fractionated extracts using 30 µl (10 mg/ml) of Dynabeads M280 slurry per condition. Dynabeads were blocked overnight at 4 °C under agitation, in blocking buffer [20 mM Hepes pH 7.9, 0.05% NP40, 150 mM NaCl, 15% Glycerol, 2 mM DTT, and 20 mg/ml BSA]. Coupling of beads with 3 µg of biotinylated nucleic acids (b-ssDNA, b-dsDNA or b-dsRNA) was subsequently performed in washing buffer [1 M NaCl, 5 mM Tris pH 7.4, 0.5 mM EDTA] for 15 min at 25 °C under gentle agitation. Nucleic acid-coupled beads were then washed twice using washing buffer and equilibrated in TENTG-150. Subsequently, 4 mg of total protein was added to the equilibrated beads and incubated at 4 °C for 3 hours in low-binding tubes (Axygen) on a wheel. Following incubation, three consecutive washes were performed with TENTG-150, changing tubes at the first and last washes. Bound proteins were eluted in 30 µl of Laemmli buffer for 5 min at 95 °C prior to WB analyses. The interaction between proteins and tested biotinylated nucleic acids was assessed by Western Blot.

### In-cell biotinylated nucleic acid pull-down

Cells were transfected with 1 to 4 µg of biotinylated synthetic nucleic acid probes using JetPrime, for up to 6 h prior to lysis in TENTG-150 for 30 min at 4 °C. Lysates were centrifuged for 30 min at 12,000 g at 4 °C prior to 45 min incubation at 4 °C with 30 µl of Dynabeads M280 blocked and equilibrated as above. Following the incubation, three washes were performed as above prior to elution of bound proteins in 30 µl Laemmli buffer and subsequent WB analyses.

### Immunoprecipitation

Endogenous immunoprecipitation was performed from 1 mg of total protein from whole cell lysates using 1 µg of MeCP2-targeting antibody or control IgG. After an overnight incubation at 4 °C on a wheel, immunoprecipitation was performed using Protein G Sepharose beads. After 3 washes in TENTG-150, the bound material was eluted in Laemmli buffer prior to WB analyses.

### Immunofluorescence and microscopy analysis

For immunofluorescence analysis of MeCP2 subcellular localization following nucleic acid challenge, $3\times10^5$ MEFs or $1\times10^6$ RAW264.7 cells were seeded onto glass coverslips prior to transfection using either 2 µg unlabelled ssDNA or dsDNA, or with 0.5 µg of Cyanin3-labelled dsDNA (Cy3-dsDNA) as described above. Six hours post transfection, cells were fixed at room temperature for 15 min in PBS containing 4% para-formaldehyde (PFA). Following fixation, cells were permeabilized in PBS containing 0.1% Triton X-100 at room temperature prior to blocking in PBS-T supplemented with 5% BSA for 30 min at room temperature. The procedure used for quantification of MeCP2 subcellular localization following infection is described in the corresponding section.

Coverslips were incubated at 37 °C for 45 min with primary antibodies in PBS-T. Primary antibodies used are: anti-MeCP2 (Cell signalling D4F3, #3456 T) used at 1:50 dilution, anti-cGas (Cell signalling D3080, #31659) used at 1:100 dilution, anti-Irf3 (Cell signalling D83B9, #4302S) used at 1:100 dilution, anti-Ranbp1 (Santa Cruz Biotechnology #sc-374352) used at 1:50 dilution, anti-dsDNA (Abcam #ab27156) used at 1:100 dilution, and anti-GFP (Abcam #ab290) used at 1:100 dilution.

Following 3 washes in PBS-T, coverslips were incubated for 30 min at 37 °C with appropriate secondary antibodies at 1:200 dilution in PBS-T. Secondary antibodies used include: Alexa Fluor 488-coupled goat anti-Rabbit IgG (Thermofischer #R37116), Alexa Fluor 488 goat anti-Mouse IgG (#A11001, Thermofischer), Alexa Fluor 594-coupled goat anti-Mouse IgG (Thermofischer #R37121), and Alexa Fluor 594-coupled goat anti-Rabbit IgG (#R37117, Thermofischer). After incubation with secondary antibody, coverslips were washed 5 times in PBS-T. Nuclei were stained with 1 µg/ml DAPI in PBS for 5 min. The coverslips were washed 5 times with PBS prior to mounting in anti-fade media (Vectashield). Adjustment of individual colour channels has been done to allow the visibility of signals. Image were taken with the Confocal Zeiss LSM980 NLO or with the Zeiss Axioimager Z3 Apotome from MRI and (Montpellier Ressources Imagerie). Images acquired with the LSM980 NLO, objective lens is a Plan-Apochromat 63x/1.40 Oil DIC M24. Detectors are Mutilalkali-PMT or GaAsP-PMT (2 PMT fluorescence, 1 multi-PMT spectral detector, 1 PMT transmission, 1 camera sCMOS); diodes for excitation are: 405 nm, 488 nm, 561 nm and 639 nm; with respective detection wavelengths 413-469 nm, 490-535 nm, 644-758 nm and 570-632 nm. Images acquired with Apotome; objective is alpha Plan-Apochromat 63x/1.46 Oil Korr M27. Detectors are one camera ORCA-Flash4 LT Hamamatsu monochrome (2048 ×2048 pixels, 6.5 µm pixel size) and one camera ZEISS quadriCCD Axiocam 506 6MP colour (2752 ×2208 pixels, 4.54 µm pixel size). Fluorescence cubes are for Hoechst, GFP, Texas Red and Cy5 with respective wavelengths 353-465 nm, 488-509 nm, 592-614 nm and 650-673 nm. Image sizes are 1024 × 1024 pixels or 2048 ×2048 pixels with Bit depth 8 bits or 16 bits. The number of images per condition varies between 5 to 20. Images were processed with Omero, Fiji or with the CellProfiler software.

### cGAMP quantification

Cells were transfected in OptiMEM with dsDNA (2 µg) for 6 hours, harvested in ice-cold PBS, counted for normalization and extracted using the Mammalian Protein Extraction Reagent (M-PER) buffer (ThermoFisher) prior to quantification using the 2'3'-cGAMP enzyme-linked immunosorbent assay (ELISA) (Cayman), according to the manufacturer's protocol.

### Proteome profiler

To assess differential chemokine expression upon dsDNA transfection, Proteome Profiler Mouse Chemokine XL Array Kit (ARY028) was used following manufacturer's instructions. Conditioned media from MEF$^{gCTRL}$ or MEF$^{gMecp2}$ cells were used.

### HSV-1 McKrae(EGFP) construction and amplification

McKrae(EGFP) was generated as described in Russell et al. (2015)[1]. In brief, CRISPR/Cas9 plasmid targeting the UL26-UL27 region and the repair plasmid backbone with UL26-UL27 flanking regions were a kind gift from David C. Tscharke. The repair plasmid pUL26/27-pICP47-EGFP encoding the HSV-1 ICP47 promoter and EGFP flanked by HSV-1 UL26 and HSV-1 UL27 (both 1100 bp) was generated by Gibson cloning (New England Biolabs). To generate recombinant McKrae(EGFP) by homologous recombination, the UL26/27-pICP47-EGFP repair plasmid was enzymatically linearized overnight and purified with PureLink PCR Purification Kit (Life Technology).

A monolayer of HEK293T cells was co-transfected with the repair plasmid and the CRISPR/Cas9 plasmid (Lipofectamine, Life Technology). 5 h post-transfection, cells were infected with McKrae 0,1 MOI. 4dpi infected cells were harvested and transferred to -80 °C. To purify McKrae(EGFP), the infected HEK293T cell suspension underwent 3 cycles of freeze-thawing followed by serial dilution on Vero cells. 0.2% IgG (Beriglobin) was added to the Vero cell media to allow virus plaque formation. EGFP-positive plaques were identified by fluorescent microscopy and transferred to 500uL DMEM(2%FCS).

To ensure absence of contamination, 3 rounds of plaque purification on Vero cells were performed.

The sequence of the EGFP insert in the final purification solution was confirmed by Sanger sequencing. Vero cells with 1 EGFP positive plaque were harvested in PCR buffer supplemented with proteinase K (NEB, Sigma) and transferred to −80 °C. Upon heat inactivation (20 min: 56 °C, 10 min: 85 °C), the solution was used in three PCR reactions to cover the pICP47-EGFP insert as well as the UL26-UL27 flanking regions.

## HSV-1 Infection

The HSV-1 KOS-64 GFP strain was a gift from S. R. Paludan (Aarhus University, Denmark), while HSV-1 McKrae(EGFP) was engineered as described above. HSV-1 viruses were amplified on Vero cells, aliquoted, and frozen at −80 °C. Titration was performed on Vero cells by serial dilutions and plaque formation assessment to determine the multiplicity of infection (MOI). For gene expression analysis, $2.5 × 10^5$ cells were seeded per well of 6-well plates 24 hours prior to infection with 5 MOI for 6 or 16 hours. Cells were subsequently harvested and RNA extracted with TRIzol for gene expression analyses. For immunofluorescence of MeCP2 subcellular localization following infection, cells were treated as above, except that cells were grown on coverslips prior to infection and fixation using 4% PFA in PBS. Image acquisition was performed using an apotome microscope (Zeiss) or with a Confocal Zeiss LSM980 NLO.

For assessment of HSV-1 infection by plaque formation assay, 1 $×10^4$ cells of control of MeCP2-targeting gRNA expressing MEFs were seeded per well of 96-well plates. Twenty-four hours after plating, cells were infected with HSV-1 at indicated MOIs in infection medium (DMEM supplemented with 1% FBS, 1% penicillin/streptomycin and 1% Glutamine) for 90 minutes before medium replacement with DMEM supplemented with 1% human serum, 1% penicillin/streptomycin and 1% Glutamine. Sixteen hours later, medium was replaced with DMEM supplemented with 10% FBS medium,1% penicillin/streptomycin and 1% Glutamine for an additional 32 hours. Cells were either fixed with 4% paraformaldehyde (PFA) prior to staining with crystal violet and plaque counting or supernatants collected to infect WT-MEF and assessment of infected cells by GFP signal quantification.

## VSV-GFP infections, quantifications and imaging

VSV-GFP was a gift from Dr Sebastien Pfeffer (IBMC, France). VSV-GFP was amplified on BHK21 cells, aliquoted and frozen at -80 °C. Single-round titration of viral stock was performed on NIH-3T3 cells in 96-well plates in technical triplicate. Briefly, cells were infected with serial dilutions of VSV-GFP for 1 h in DMEM 2% FBS, washed with PBS and media replaced. Cells were fixed using 4% PFA for 20 mins at room temperature, permeabilized in 0.2% Triton X-100 for 10 min and nuclei stained with DAPI. The percentage of infected cells (i.e., GFP-positive cells) was determined using an ImageXpress Pico (Molecular Devices) and the number of infectious particles per ml was determined. To determine the infectivity of VSV-GFP in RAW cells expressing control or *Mecp2*-targeting gRNAs, the same protocol as above for the NIH-3T3 titration was used.

For confocal imaging experiments, RAW264.7^CTRL or RAW264.7^Mecp2-/- were plated in either 96-well plates ($6 × 10^5$ cells) or in 24-well plates on glass coverslips ($4×10^6$ cells) and infected the next day with the indicated MOIs of VSV-GFP for 7h30 and fixed as above. Cells were permeabilized as above, blocked with 10% Normal Goat Serum (Invitrogen) for 1 h and immunofluorescence was performed. For cells in 96-well plates, the cells were treated as above. For cells on glass coverslips, cells were incubated with primary antibodies (Rabbit anti-MeCP2, Cell Signalling Technologies, D4F3, 1/50 and Mouse anti-VSVG, Kerafast, Kf-Ab01401-2.0, 1/250) in PBS 1supplemented with 0.5% BSA for 1 h followed by incubation with Alexa Fluor secondary antibodies

(Thermo Fisher Scientific) and DAPI for 1 h. Cells were washed in water and mounted on slides using ProLong Gold Antifade Mountant (Thermo Fisher Scientific). Imaging was performed on an LSM980 8Y confocal microscope (Zeiss), using a 63x lens for glass coverslips and a 10x lens for 96-well plates. Post-processing was performed using the FIJI software[51].

## RNA-Seq data analysis

Raw reads were subjected to quality control and adapter trimming using Trimgalore[52] (version 0.6.6). Trimmed reads were then aligned to the reference genome (mm9) using HISAT2 (version 2.2.1)[53] Converting SAM to BAM, sorting and indexing BAM files was performed using samtools[54] (version 1.11). Aligned sequencing reads have been counted using the HTSeq tool[55] (version 0.11.3; options: -i gene_id --additional-attr gene_name -t exon). Differentially expressed genes were detected using DESeq2 R package (version 1.32.0)[56] Bigwig files were generated using the bamCoverage tool from the deepTools suite[57] (version 3.4.2; options: --smoothLength 15 --normalizeUsing RPKM). For transposable element expression (retrotranscriptome) analyses, trimmed reads were mapped to the reference genome (mm9) using HISAT2 (version 2.2.1)[53]. The aligned reads were converted from SAM to BAM format, sorted, and indexed using Samtools (version 1.11). Aligned reads were then counted using Telescope (version 1.0.3.1)[58].

## Cytosolic DNA extraction

Cytosolic DNA extraction was performed using a protocol adapted from ref. 59. In brief, cytosolic DNA was isolated from cultured cells kept on ice and washed prior to collection by low-speed centrifugation (500×g for 10 minutes at 4 °C) to obtain a compact pellet. Cell pellets were resuspended in a Cytosolic Extraction Buffer (5 M NaCl, 1 M HEPES, 20 mg/mL Digitonin, 98% Hexylene Glycol, ddH$_2$O), optimized for each cell line to selectively disrupt the plasma membrane while preserving nuclei and organelles. A second low-speed centrifugation (2000×g for 10 minutes at 4 °C) was performed to separate the cytosolic and nuclear fractions and the supernatant was collected as the cytosolic fraction, and the pellet as the nuclear fraction. Fractions were verified by Western blot. The cytosolic fraction was further clarified by high-speed centrifugation (18,300×g for 10 minutes at 4 °C) to remove residual particulates. It was then treated with Proteinase K, followed by RNase and phenol-chloroform-isoamyl alcohol.

## MinION sequencing data

Cytosolic DNA was extracted as described above. Nanopore sequencing (Oxford Nanopore Technologies) was performed using a Ligation sequencing kit V14 (SQK-LSK114) and the Native barcoding kit 24 (OXNTSQK-NBD114). The sequencing library was loaded onto a Flow cell R10.4.01 (FLO-MIN114) on a MinION Mk1B sequencer. MinKNOW (version 24.02.8, ONT) was used for sequence planning and data acquisition, with the super-accurate basecalling mode enabled. Dorado (version 7.3.11) was used as the basecalling algorithm to produce pod5 and fastQ files. The automatic real-time division into passed and failed reads by the MinKNOW was used as a quality check, removing reads with quality scores <10. The quality-checked reads were demultiplexed and trimmed for adapters and primers using Dorado, followed by mapping with the reference genome (mm9) using Minimap2[35] (version 2.28) and then converted from SAM to BAM format, sorted, and indexed using Samtools (version 1.20). The aligned reads were then counted using Telescope (version 1.0.3).

## Analyses of public datasets

To test whether MeCP2 is involved in innate immune response and inflammation, we extracted the list of genes for the Gene Ontology

(GO) terms with the keywords "inflammation", "interferon-beta", "viral infection", "sting", and "innate immunity" in the description. We then used EdgeR to extract the list of significantly differentially expressed Genes (DEGs) in studies comparing samples where the MeCP2 function was disrupted, in human and animal studies and we tested whether the genes involved in the groups listed above were present among the DEGs. For the study in humans, we considered the samples taken from patients with RTT with different severity of symptoms ([36] - GEO accession: GSE198856). For the animal study, we used the study of[35] – GEO Accession: GSE107004), where the authors used RNA interference to disrupt the MeCP2 expression in the hippocampus of adult mice. In all the cases considered, we found that among the DEGs there are genes involved in the groups listed above. For analyses of gene set enrichment, we used R project, and the full list of DEGs was used to derive significant gene sets in the Gene Set Enrichment Analysis (GSEA). We report the top ten list of genes based on the Normalized Enriched Score (NES). These findings are consistent with the state of inflammation detected in patients with RTT[37].

## Image quantification and statistical analysis

Quantification of the cytosolic and nuclear intensity of MeCP2 was performed using the Fiji or CellProfiler softwares. Co-localization analysis was conducted utilizing the JACoP plugin and colocalization test within ImageJ (Fiji).

Statistical analyses were performed using GraphPad Prism version 9 or the EdgeR software. A bi-directional unpaired t-test was performed except when otherwise stated. For correlation analysis, Pearson's correlation coefficient was used for normally distributed data. The number of independent experimental replicates for each experiment is provided in the figure legends. All data are expressed as mean ± standard error of the mean (SEM). N.S.: non-significant. $*P < 0.05$, $**P < 0.01$, $***P < 0.001$ and $****P < 0.0001$.

## Reporting summary

Further information on research design is available in the Nature Portfolio Reporting Summary linked to this article.

## Data availability

The sequencing data that support the findings of this study are available in Gene Expression Omnibus (GEO) with the accession number: GSE286551. Other Source data are provided with this paper.

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

## Acknowledgements

We thank all members of the Molecular Basis of Inflammation laboratory for their critical reading of this manuscript. We thank Denis Tempe for discussions. We thank Soren Paludan for WT and cGas-deficient MEF cell lines and HSV-1-GFP, J Rewhinkel for Trex1-deficient MEFs, Olivier Moncorgé for BHK21 cells and Sébastien Pfeffer for VSV-GFP. We acknowledge the MRI imaging facility, a member of the national infrastructure France-BioImaging infrastructure supported by the French National Research Agency (ANR-10-INBS-04, "Investments for the future"). We thank Julian Venables for the edition of the manuscript. This work was co-funded by the European Union (ERC, SENTINEL 101087092 to NL and DELV 101039538 to KM). Views and opinions expressed are however, those of the author(s) only and do not necessarily reflect those of the European Union or the European Research Council. Neither the European Union nor the granting authority can be held responsible for them. This work was also co-funded by LA LIGUE pour la recherche contre le cancer [AAPARN 2021.LCC/JuF (NL) and PhD fellowship (HC)], the Agence Nationale de Recherche sur le SIDA et les Hépatites virales (ANRS) [ECTZ117448 (NL)], the AFSR (Association Française du Syndrome du Rett) (EV, ZH), the I-SITE Excellence Programme of the University of Montpellier, under the Investissements France 2030 [RETTiNA (NL), ChoiCe (NL) and PhD Fellowship (SG)], the Fondation ARC [ARCPJA2021060003720 COPALYS (NL) and PhD fellowship (HC)], La Région Languedoc Roussillon [Prématuration 2021 MODULON 21015964 (NL)] and the Centre National de La Recherche Scientifique [Prématuration CNRS (NL)]. Research in the Tropea Lab has emanated from research supported in part by a research grant from Research Ireland under Grant Number 21/RC/10294_P2 (DT) and co-funded under the European Regional Development Fund (DT) and by FutureNeuro industry partners (DT), and from the grant #3507 from the International Rett Syndrome Foundation (IRSF) (DT). Work in SRP and FIS lab was funded by: Independent Research Fund Denmark DFF-International Postdoc 8026-00014B (SRP and FIS) and Danish National Research Foundation (DNRF164) (SRP and FIS).

## Author contributions

Conceptualization: N.L.; Methodology: H.C., I.K.V., S.G., C.A.S., M Schüssler, N.L.; Investigation: H.C., I.K.V., A.C., S.G., C.T., R.C., C.A.S., M. Salma, M.A.D., M.C., M. Schüssler, Z.H., J.M., P.L.H., Y.M.N., R.J.E., A.A., J.R., M.H.C., D.T.; Visualization: H.C., N.L., I.K.V., M. Salma, C.T., P.L.H., D.T.; Funding acquisition: N.L.; Project administration: N.L.; Supervision: I.K.V., N.L., E.S., E.V., K.M., S.R.P., F.I.S.; Writing – original draft: N.L.; Writing – review & editing: H.C., S.G., I.K.V., N.L., M. Schüssler, C.T., C.A.S., E.S., M. Salma, E.V., J.M., K.M., D.T.

## Competing interests

The authors declare no competing interests.
