## [Transparent Peer Review file · Nature Communications]

The methyl-CpG-binding protein 2 inhibits cGAS-associated signaling

Corresponding Author: Dr Nadine Laguette

Version 0:

Reviewer comments:

Reviewer #1

(Remarks to the Author)

[Editorial Note: See attachment at end of file]

Reviewer #2

(Remarks to the Author)

This manuscript by Chamma et al explores a potential role for MeCP2 in regulating cytoplasmic dsDNA/cGAS-mediated inflammatory responses. The authors report that cytoplasmic dsDNA causes MeCP2 to accumulate in the cytoplasm where it then moderates inflammatory responses by competing with cGAS for dsDNA binding. As the authors point out, MeCP2 is well-characterized as a nuclear protein. Their findings would therefore be interesting (and surprising) to those researching the biology of Rett syndrome, and also to those studying inflammation.

The evidence presented in the manuscript as it stands, however, does not substantiate the authors conclusion that MeCP2 is present in the cytoplasm, and that it functions there to moderate inflammatory signalling.

Major points

A key issue is the specificity of the antibody used to detect MeCP2 in the cytoplasm following dsDNA challenge. An essential control is to use MeCP2 null cells for these experiments. This would verify that the cytoplasmic signal attributed to MeCP2 does indeed correspond to the presence of cytoplasmic MeCP2. Ideally a co-culture of wild-type and MeCP2 null cells challenged with dsDNA would only show cytoplasmic MeCP2 signal in the wild-type cells. Live imaging experiments with GFP-MeCP2 knock-in cells (and untagged cells as a control) would also be desirable to confirm the central observation of cytoplasmic MeCP2.

The authors report that MeCP2 deficiency increases IFN β expression in dsDNA-challenged cells by approx 10-fold (Fig 4C). However, gene expression changes seen in MeCP2 deficient cells/tissue are notoriously subtle and difficult to pin down. Is this deregulated gene expression seen across multiple clones of MeCP2 WT vs. KO cells? Can the upregulation IFN β in MeCP2 KO cells be rescued by re-expression of exogenous MeCP2?

The presentation of the RNA-seq data is confusing and the conclusions drawn from these experiments are over-interpretation. It is claimed that "displacement of MeCP2 from the nucleus disrupts its canonical function" – but the experiment underlying this statement is RNA-seq demonstrating induction of retroelements upon challenge with dsDNA. Substantiation of this conclusion would depend on observing a similar level of induction of retroelements upon deletion of MeCP2 in the same cell type (RAW264.7). Further, if dsDNA-mediated induction of retroelements is due to displacement of MeCP2 from the nucleus, the dsDNA should not affect retroelement transcription in MeCP2 null cells.

Minor points

The title of the figure 3 legend has a spelling mistake (increases).

Figure 5B has essentially every cell infected with HSV at a MOI of 0.5.

Figures S5 and S6 are labelled as Figures 5 and 6.

Were microscope settings the same for all images acquired? In Fig S2B, the reported appearance of MeCP2 in the

cytoplasm after dsDNA challenge is accompanied by a striking increase in nuclear MeCP2 signal. In one experiment (Fig 2A) dsDNA causes lamin to disappear from the cytoplasm. In another experiment (Fig S2A) dsRNA causes lamin to appear in the cytosol. How do the authors explain this difference in behaviour? The authors report altered expression of IFN/antiviral response genes in MeCP2 null livers. Are these genes also found to be dysregulated in MeCP2 null brain/neurons (in the many published datasets)? If these findings are relevant to the neuropathology of Rett syndrome, then a prediction is that these genes should also be misregulated in MeCP2-deficient neural tissue.

Version 1:

Reviewer comments:

Reviewer #1

(Remarks to the Author)

All of my concerns have been addressed.

Reviewer #2

(Remarks to the Author)

The authors have addressed some of the issues raised by myself and the other reviewers. I remain concerned about the specificity of the MeCP2 antibody used for detection of MeCP2 in the cytoplasm (a critical reagent for many of the experiments supporting the main conclusions of this manuscript). The data in the manuscript (Fig S3A and S4A/B) nicely shows that the antibody used specifically detects nuclear MeCP2. However, the only evidence that the cytoplasmic MeCP2 signal is absent in MeCP2 KO cells is the plot in Figure B from the rebuttal letter, and we are not shown the images which are summarized in this plot. This data should be presented to the readers/reviewers to allow them to make up their minds about the quality/reliability of this key experiment. Optimally, a co-culture experiment of wild-type and MeCP2 KO cells treated with dsDNA would put to rest any concerns about artefacts and batch effects in this experiment.

Version 2:

Reviewer comments:

Reviewer #2

(Remarks to the Author)

The manuscript is now suitable for publication.

made.

Reviewer #1 (Remarks to the Author):

In this manuscript, Chamma et al. found that the methyl-CpG-binding protein 2 (MECP2), a major transcriptional regulator, controls dsDNA-associated inflammatory responses. The authors showed that the presence of cytosolic dsDNA promotes MECP2 export from the nucleus to the cytosol where it interacts with dsDNA, dampening cGAS activation. Further, they showed that the absence of Mcep2 enforces an antiviral state. Finally, they found that the displacement of Mcep2 from the nucleus following dsDNA stimulation disrupts its canonical function. This study provides a novel role and regulatory mechanism of Mcep2 in dsDNA-associated inflammatory responses. The manuscript are well written, it is easy for the reader to understand. However, the quality of most of figures need to be improved, several key experiments need to be performed to provide more mechanistic insights into Mcep2-regulated antiviral innate immune responses.

We thank the reviewer for critical assessment of our manuscript. To improve the quality of the data we provide, deepen mechanistic insights, and improve readability, we reorganized figures 1 and 2 and we addressed the raised concerns, as detailed in the point-by-point answers below:

Major comments:

1. Figure 1A, the authors claimed that Mcep2 selectively interacts with dsDNA but not ssDNA. However, Mcep2 also binds to ssDNA even though this binding is weak. The authors should state results rigorously. Additionally, the authors should detect the binding of Mcep2 to ssDNA in Figure 1B.

The text has been modified to state the results more accurately. ssDNAs were not used in original Fig. 1b. We have now provided data where ssDNAs and dsDNA are used for in vitro pull-down experiments in RAW264.7 cells. This data (new Fig. 1c) shows that in RAW264.7, the binding of Mecp2 and cGas is weaker in the ssDNA condition as compared to the dsDNA condition.

2. Figure 1C and 1D, do authors explain why the protein levels of Mcep2 or cGas are so low in cells that have not been transfected with dsDNA (Input samples)? The expression of Mcep2 is induced by dsDNA stimulation? The low protein levels of cGas in un-infected/transfected cell lines seem unusual.

The input levels of Mecp2 appear to be low, but this is because:

- 1- we used exposures that allow the pulled-down signals to no be saturated. We have now added longer exposures that allow better visualization of Mecp2 and cGas in the Mock-transfected input samples.
- 2- cGas is an interferon stimulated gene. As can be seen in the panels added to Supplementary Fig. 1b, upregulation of cGas is visible at the mRNA level upon dsDNA transfection.
- 3- To the contrary, Mecp2 is not reported to be an interferon stimulated gene, and its mRNA levels do not change upon dsDNA stimulation (see panel added to Supplementary Fig. 1b).

The apparent low levels of Mecip2 and cGas that can be seen by WB are also linked to the extraction buffer that is used to perform most of the experiments presented in the manuscript, which is mild (150mM salt). This buffer was optimized for compatibility with the pull-downs we perform. Figure A (left) presents the levels of proteins extracted using this buffer, as compared to extraction with RIPA buffer. Together with qPCR data, these WB analyses suggest that Mecip2 is stabilized following dsDNA transfection.

Figure A: WT-MEF were transfected or not with dsDNA prior to whole cell extract preparation using TENTG 150 or RIPA buffer. WB was performed using indicated antibodies*: longer exposure.

3. Figure 4-6, the authors only measured transcription levels of *Ifn- α* , *Il-6*, and *Cxcl10* after dsDNA transfection or HSV-1 infection. The authors should examine the protein levels of these downstream genes by ELISA assay. The phosphorylation of TBK1, STING, and IRF3 should also be examined by immunoblotting.

To address this concern, we have now included additional inflammatory genes in several panels (Fig 4c, 4f, 4g, 5e, and Supplementary Fig. 4c), WB analyses of pathway activation (Supplementary Fig. 4d), as well as proteome profiler-based cytokine measurement (Fig. 4d-e).

4. Mecip2 specifically binds to DNA? How about dsRNA? The presence of dsRNA promotes Mecip2 export? Does Mecip2 involve in RNA virus-triggered signaling?

To address these questions, we now provide:

- Pulldown and immunofluorescence experiments performed using dsRNA (new Fig 1d and Supplementary Fig. 1g)
- Experiments where control and Mecip-2 deficient MEFs are challenged with poly(I:C) (Fig. 4h).

These experiments show that (1) Mecip2 does not bind dsRNA and (2) absence of Mecip2 does not alter poly(I:C) associated inflammatory responses.

Importantly, our study does not rule out that RNA viruses may also exploit Mecip2 in the context of their life cycle. Using VSV as a model RNA virus, we show that the presence or absence of Mecip2 does not affect VSV infection (Supplementary Fig. 5b and 5d), suggesting that RNA viruses may bypass its antiviral effects. We now discuss this more in our manuscript.

5. In the entire manuscript, the authors only performed experiments using the dsDNA indicated in methods. The authors also claimed that the absence of Mecip2 enforces an antiviral state. How about other dsDNA transfection? HSV60? HSV120? Other DNA virus? VACV? HCMV?

To address these questions, we now provide experiments performed with:

- Another dsDNA sequence (see methods), used in pulldown analyses (Fig. 1e) and Immunofluorescence analyses (Supplementary Fig. 1h)
- Infections performed with HSV-McKrae in addition to KOS64 (Fig. 5d and Fig. S6h-i)

Experiments were performed in cells proficient or deficient for Mecip2 and confirm that the enhanced inflammatory responses observed in Mecip2 deficient cells in response to dsDNA is independent of the DNA sequence used.

6. Figure 3C, the interaction between Mecip2 and cGas is so weak. It would be better to perform this experiment in MEF cells transfected with dsDNA for different time points. As Figure 3C showed that the expression of cGas is significantly induced after dsDNA transfection, so the authors should examine the interaction of other proteins which are also induced by dsDNA stimulation with Mecip2 as a negative control.

To address this point, we provide new Fig. 3e, where Flag-tagged Mecip2 is stably expressed in Mecip2 ko cells prior to transfection of dsDNA for 1, 3, 6 and 16h prior to FLAG immunoprecipitation, coupled to analysis of co-immunoprecipitation of cGas. Isg15 is used as a control of protein induced by dsDNA transfection. While we found that over time the interaction of Mecip2 and cGAS is increased by the presence of dsDNA, Isg15 is induced by dsDNA transfection in input material, but does not co-IP with Mecip2.

7. Figure 3D, the absence of Mecip2 increases the expression of cGas? The authors claimed that the reduced levels of Mecip2 led to increased cGas recruitment to dsDNA. Actually, the authors hardly made this conclusion because of the induced expression of cGas in Mecip2 knockout cells. The authors should clarify whether Mecip2 regulates the expression of cGas.

Experiments conducted in Figure 3 show that in vitro, there is enhanced recruitment of cGas to dsDNA species in absence of Mecip2. There is no consistent increase of cGas upon Mecip2 knockout. We apologize for this confusion in the previous text and hope that the new Figures 3 and 4 make the point that absence of Mecip2 leads to enhanced cGas-dependent signaling.

8. As Figure 3E showed that the absence of cGas increases Mecip2 interaction with cytosolic dsDNA. It would be better to assess Mecip2 subcellular localization in MEF *Sting*^{-/-} cells in Figure 2F.

Our recent work (and other recent publications), highlight that STING plays multifunctional roles in the maintenance of homeostasis. We hence avoided the use of STING-deficient cells in functional experiments for metabolic alterations may impact on several proteins. However, to address the raised concern, we now provide pulldown experiments performed in STING KO cells (See Supplementary Fig. 3c).

9. Figure 3E, the author present MEFcGas^{-/-} mistakenly in input samples. The authors should check this carefully.

This is a labeling mistake that has been corrected.

10. The authors claimed that Mecip2 and cGas interact with the same dsDNA moieties in the cytosol and the absence of Mecip2 enforces an antiviral state. Do Mecip2 and cGas bind to the overlapping regions of HSV-1 genome?

Indeed, it is possible that Mecip2 and cGas bind to overlapping regions of HSV-1 genome. This possibility is now discussed in the manuscript.

Minor comments:

1. The authors need to check the manuscript and Figures carefully, “RAW264.7 cell line” is correct but not “RAW246.7 cell line”.

We carefully checked the manuscript and corrected this typo throughout.

2. How do authors generate MEFcGas^{-/-} or MEF^{Trex1}^{-/-} cells? Those cells were obtained from knockout mice? If they are, the authors should provide the information of knockout mice.

MEF^{cGas}^{-/-} were a gift from SR Paludan while MEF^{Trex1}^{-/-} were obtained from the Rewinkel lab. This is now clarified in the materials and methods section.

3. The authors should present the data clearly. Figure 1A (left) showed that the representative of “Mock” is streptavidin beads-without bound DNA, whereas the representative of “Mock” in Figure 1C (left) is cell line without dsDNA transfection. To clarify the Figures, the authors should differentiate between them.

This is now clarified in the Figure legend.

4. P.10, line 352, “interact” should be singular.

This typo has been corrected.

5. P.10, line 363, the appearance of “k,l” in the sentence seems unusual, what do those letters refer to?

This typo has been corrected.

6. According to the statement in P. 12, Line 447-451, the order of Figure 3D and Figure 3E in Figure 3 could be wrong. Otherwise, the statement of results is not consistent with the presence of Figures. The authors should check this carefully.

This has been corrected.

7. As the authors stated in P. 13, line 457, it would be better to provide the reference here.

Reference has been provided.

8. P. 15, line 538, “MeCP2” or “Mecip2”? Which is correct?

The spelling of Mecip2 has been checked throughout. For general statements or when the human protein is mentioned, we used ‘MECP2’; when the Mouse protein is mentioned, we used ‘MeCP2’; when referring to the Mouse gene: *Mecip2*.

9. Figure 4D, the authors should also examine the transcription levels of indicated genes upon dsDNA stimulation in MEFgCTRL cells treated with H-151.

This is now provided in Figure 4f.

10. Figure 4E, the expression of FLAG-Mecp2 should be monitored by immunoblotting.

Protein levels of FLAG-Mecp2 are now provided as Supplementary Fig. 4g.

11. Figure S5C, MEFgCTRL and MEFgMecp2 should be presented.

Previous Supplementary Fig. 5C, now included as Fig. 5g-h, presents conditioned media experiments where virus containing supernatants from MEFgCTRL and MEFgMecp2 were used to infect WT-MEF. Consequently, only WT-MEFs are shown. This is now clarified in the legend and text.

12. In the manuscript (e.g P. 13, line 466), the authors only examine the transcription levels of genes classically associated with cGas activation by qPCR assay but not ELISA assay. So the authors should state the results rigorously.

We both more rigorously state our results, but also provide cytokine measurements (Fig. 4d-e) to reinforce our message.

Reviewer #2 (Remarks to the Author):

This manuscript by Chamma et al explores a potential role for MeCP2 in regulating cytoplasmic dsDNA/cGAS-mediated inflammatory responses. The authors report that cytoplasmic dsDNA causes MeCP2 to accumulate in the cytoplasm where it then moderates inflammatory responses by competing with cGAS for dsDNA binding. As the authors point out, MeCP2 is well-characterized as a nuclear protein. Their findings would therefore be interesting (and surprising) to those researching the biology of Rett syndrome, and also to those studying inflammation. The evidence presented in the manuscript as it stands, however, does not substantiate the authors conclusion that MeCP2 is present in the cytoplasm, and that it functions there to moderate inflammatory signalling.

Major points

1. A key issue is the specificity of the antibody used to detect MeCP2 in the cytoplasm following dsDNA challenge. An essential control is to use MeCP2 null cells for these experiments. This would verify that the cytoplasmic signal attributed to MeCP2 does indeed correspond to the presence of cytoplasmic MeCP2. Ideally a co-culture of wild-type and MeCP2 null cells challenged with dsDNA would only show cytoplasmic MeCP2 signal in the wild-type cells. Live imaging experiments with GFP-MeCP2 knock-in cells (and untagged cells as a control) would also be desirable to confirm the central observation of cytoplasmic MeCP2.

To address this comment, we have now included experiments where the Mecip2 antibody is tested in control and Mecip2 null cells (MEF and RAW264.7) (see Supplementary Fig. 3a and Supplementary Fig. 4a-b).

We also performed experiments where we stained Mecip2 at several time points following dsDNA stimulation, showing that Mecip2 cytosolic staining is increased over time following dsDNA stimulation, accompanied by decreased nuclear staining, with a peak at 6 hours (Fig. 2a-b).

Although not included in the manuscript, we also performed this time-course experiment in Mecip2 knockout cells, as well as Mecip2 cytosolic foci quantification. Figure B below shows that the increase of Mecip2 cytosolic signal observed in control cells is not observed in Mecip2 KO cells at any of the tested time points.

Figure B: Quantification of Mecip2 cytosolic foci in Mecip2 proficient and deficient cells.

2. The authors report that MeCP2 deficiency increases IFN β expression in dsDNA-challenged cells by approx 10-fold (Fig 4C). However, gene expression changes seen in MeCP2 deficient cells/tissue are notoriously subtle and difficult to pin down. Is this deregulated gene expression seen across multiple clones of MeCP2 WT vs. KO cells?

Can the upregulation IFN β in MeCP2 KO cells be rescued by re-expression of exogenous MeCP2?

We agree that MECP2-deficiency-associated gene expression changes are hard to point down. However, the type I IFN signature was observed in several cellular contexts. Inflammatory responses were assessed on pools of Mecp2 KO cells and in two cell lines (not in isolated clones), in order to avoid clonal effects. To further support that this can be seen across various cellular contexts, we now include data obtained using another gRNA (see supplementary Fig. S4e-f).

In addition, we already reported a signature of increased type I IFN responses in mice liver (Fig. 5j-k) and now included analyses of public dataset from mice (Fig. 5i and supplementary Fig. 5e-f) as well as patient datasets (Fig. 5m and supplementary Fig. 5g-h), all displaying IFN signatures.

Despite attempting several strategies for a rescue experiment, we did not succeed, likely due to technical issues, such as Sting protein levels being affected by plasmid transfection and retroviral transduction in the Mecp2-deficient background.

3. The presentation of the RNA-seq data is confusing and the conclusions drawn from these experiments are over-interpretation. It is claimed that “displacement of MeCP2 from the nucleus disrupts its canonical function” – but the experiment underlying this statement is RNA-seq demonstrating induction of retroelements upon challenge with dsDNA. Substantiation of this conclusion would depend on observing a similar level of induction of retroelements upon deletion of MeCP2 in the same cell type (RAW264.7). Further, if dsDNA-mediated induction of retroelements is due to displacement of MeCP2 from the nucleus, the dsDNA should not affect retroelement transcription in MeCP2 null cells.

To address this concern, we now include RNAseq data obtained in RAW 264.7 cells where LINE-1 upregulation is compared between control and Mecp2-deficient RAW264.7 cells in the presence or absence of dsDNA stimulation (Fig. 6e). This showed indeed that Mecp2 deficient cells present increased basal LINE1-derived transcripts, which are less upregulated upon dsDNA stimulation in KO cells versus control cells.

In addition, to reinforce our statement, we also show that LINE-1 upregulation can be measured also in WT-MEF (Figure 6d, supplementary Fig. 6d-e). In addition, Orf1p protein levels were also assessed in RAW264.7 upon dsDNA transfection (supplementary Fig. 6f-g). Finally, increased Orf1p were also observed following Mecp2 knockout, increased by HSV-1 infection in control cells and very marginally boosted in Mecp2 ko cells (fold 1.2) (supplementary Fig. 6h-i).

Minor points

The title of the figure 3 legend has a spelling mistake (increases).

This typo has been corrected.

Figure 5B has essentially every cell infected with HSV at a MOI of 0.5.

Yes, MOI are calculated on Vero cells as described in the methods of the manuscript, and the MOI leading to about half of Vero cells to be infected leads to about all MEF cells to be infected.

Figures S5 and S6 are labelled as Figures 5 and 6.

This has been corrected.

Were microscope settings the same for all images acquired? In Fig S2B, the reported appearance of MeCP2 in the cytoplasm after dsDNA challenge is accompanied by a striking increase in nuclear MeCP2 signal.

Yes, microscope settings were the same for all acquired images in a given panel, but not necessarily between panels. The difference in intensity between different panels are essentially transfection efficiency and cell type-dependent.

In one experiment (Fig 2A) dsDNA causes lamin to disappear from the cytoplasm. In another experiment (Fig S2A) dsRNA causes lamin to appear in the cytosol. How do the authors explain this difference in behaviour?

We apologize, the images that were provided were not well cropped. We now provide corrected panels (supplementary Fig. 2a-b), accompanied by the corresponding raw data.

The authors report altered expression of IFN/antiviral response genes in MeCP2 null livers. Are these genes also found to be dysregulated in MeCP2 null brain/neurons (in the many published datasets)? If these findings are relevant to the neuropathology of Rett syndrome, then a prediction is that these genes should also be misregulated in MeCP2-deficient neural tissue.

This is now included more intensively in the discussion, but in addition, we now provide analyses of public datasets from mice (Fig. 5j and supplementary Fig. 5e-f) as well as patient datasets (Fig. 5m and S5g-h), all displaying IFN signatures. It is important to point out that the expression of STING in terminally differentiated cells is controversial and the implications in neural tissue may be difficult to assess.

We thank the reviewers for their reassessment of our manuscript. We have now addressed the remaining concern of reviewer #2. Changes are highlighted in green in the manuscript. Below is the answer to the raised points.

REVIEWER COMMENTS

Reviewer #1 (Remarks to the Author):

All of my concerns have been addressed.

Reviewer #2 (Remarks to the Author):

The authors have addressed some of the issues raised by myself and the other reviewers. I remain concerned about the specificity of the MeCP2 antibody used for detection of MeCP2 in the cytoplasm (a critical reagent for many of the experiments supporting the main conclusions of this manuscript). The data in the manuscript (Fig S3A and S4A/B) nicely shows that the antibody used specifically detects nuclear MeCP2. However, the only evidence that the cytoplasmic MeCP2 signal is absent in MeCP2 KO cells is the plot in Figure B from the rebuttal letter, and we are not shown the images which are summarized in this plot. This data should be presented to the readers/reviewers to allow them to make up their minds about the quality/reliability of this key experiment. Optimally, a co-culture experiment of wild-type and MeCP2 KO cells treated with dsDNA would put to rest any concerns about artefacts and batch effects in this experiment.

We thank the reviewer for precising that the data obtained on Mecp2 ko cells upon dsDNA stimulation should be presented in the manuscript. These data are now provided as new supplementary Figure 2 panels a-b. This shows that cytosolic MeCP2 staining can only be observed in MeCP2-proficient cells upon dsDNA stimulation and not in MeCP2 knockout cells.

In addition, we have performed co-culture experiments that are presented in Supplementary Figure 2c. We did not use a cell tracker to distinguish between the MeCP2 knockout and control cells because the dyes we tested appeared to alter general cell behavior, leading to unreliable data. However, we provide images where MeCP2 proficient and deficient cells are present side-by-side. In this context, it can be appreciated that MeCP2 cytosolic staining is visible in MeCP2-proficient cells only.

Dear Nature Communications editor and colleagues, thank you for your invitation for reviewing this manuscript. My comments for this manuscript were detailed as following:

Remarks to the Author:

In this manuscript, Chamma et al. found that the methyl-CpG-binding protein 2 (MECP2), a major transcriptional regulator, controls dsDNA-associated inflammatory responses. The authors showed that the presence of cytosolic dsDNA promotes MECP2 export from the nucleus to the cytosol where it interacts with dsDNA, dampening cGAS activation. Further, they showed that the absence of Mcep2 enforces an antiviral state. Finally, they found that the displacement of Mcep2 from the nucleus following dsDNA stimulation disrupts its canonical function. This study provides a novel role and regulatory mechanism of Mcep2 in dsDNA-associated inflammatory responses. The manuscript are well written, it is easy for the reader to understand. However, the quality of most of figures need to be improved, several key experiments need to be performed to provide more mechanistic insights into Mcep2-regulated antiviral innate immune responses.

Major comments:

1. Figure 1A, the authors claimed that Mcep2 selectively interacts with dsDNA but not ssDNA. However, Mcep2 also binds to ssDNA even though this binding is weak. The authors should state results rigorously. Additionally, the authors should detect the binding of Mcep2 to ssDNA in Figure 1B.
2. Figure 1C and 1D, do authors explain why the protein levels of Mcep2 or cGas are so low in cells that have not been transfected with dsDNA (Input samples)? The expression of Mcep2 is induced by dsDNA stimulation? The low protein levels of cGas in uninfected/transfected cell lines seem unusual.
3. Figure 4-6, the authors only measured transcription levels of *Ifn- β* , *Il-6*, and *Cxcl10* after dsDNA transfection or HSV-1 infection. The authors should examine the protein levels of these downstream genes by ELISA assay. The phosphorylation of TBK1, STING, and IRF3 should also be examined by immunoblotting.
4. Mcep2 specifically binds to DNA? How about dsRNA? The presence of dsRNA promotes Mcep2 export? Does Mcep2 involve in RNA virus-triggered signaling?

5. In the entire manuscript, the authors only performed experiments using the dsDNA indicated in methods. The authors also claimed that the absence of Mcep2 enforces an antiviral state. How about other dsDNA transfection? HSV60? HSV120? Other DNA virus? VACV? HCMV?
6. Figure 3C, the interaction between Mcep2 and cGas is so weak. It would be better to perform this experiment in MEF cells transfected with dsDNA for different time points. As Figure 3C showed that the expression of cGas is significantly induced after dsDNA transfection, so the authors should examine the interaction of other proteins which are also induced by dsDNA stimulation with Mcep2 as a negative control.
7. Figure 3D, the absence of Mcep2 increases the expression of cGas? The authors claimed that the reduced levels of Mcep2 led to increased cGas recruitment to dsDNA. Actually, the authors hardly made this conclusion because of the induced expression of cGas in Mcep2 knockout cells. The authors should clarify whether Mcep2 regulates the expression of cGas.
8. As Figure 3E showed that the absence of cGas increases Mcep2 interaction with cytosolic dsDNA. It would be better to assess Mcep2 subcellular localization in MEF *Sting*^{-/-} cells in Figure 2F.
9. Figure 3E, the author present MEF^{cGas^{-/-}} mistakenly in input samples. The authors should check this carefully.
10. The authors claimed that Mcep2 and cGas interact with the same dsDNA moieties in the cytosol and the absence of Mcep2 enforces an antiviral state. Do Mcep2 and cGas bind to the overlapping regions of HSV-1 genome?

Minor comments:

1. The authors need to check the manuscript and Figures carefully, “RAW264.7 cell line” is correct but not “RAW246.7 cell line”.
2. How do authors generate MEF^{cGas^{-/-}} or MEF^{Trex1^{-/-}} cells? Those cells were obtained from knockout mice? If they are, the authors should provide the information of knockout mice.
3. The authors should present the data clearly. Figure 1A (left) showed that the representative of “Mock” is streptavidin beads-without bound DNA, whereas the representative of “Mock” in Figure 1C (left) is cell line without dsDNA transfection. To clarify the Figures, the authors should differentiate between them.
4. P.10, line 352, “interact” should be singular.

5. P.10, line 363, the appearance of “k,l” in the sentence seems unusual, what do those letters refer to?
6. According to the statement in P. 12, Line 447-451, the order of Figure 3D and Figure 3E in Figure 3 could be wrong. Otherwise, the statement of results is not consistent with the presence of Figures. The authors should check this carefully.
7. As the authors stated in P. 13, line 457, it would be better to provide the reference here.
8. P. 15, line 538, “MeCP2” or “Mecp2 “? Which is correct?
9. Figure 4D, the authors should also examine the transcription levels of indicated genes upon dsDNA stimulation in MEF^{gCTRL} cells treated with H-151.
10. Figure 4E, the expression of FLAG-Mecp2 should be monitored by immunoblotting.
11. Figure S5C, MEF^{gCTRL} and MEF^{gMecp2} should be presented.
12. In the manuscript (e.g P. 13, line 466), the authors only examine the transcription levels of genes classically associated with cGas activation by qPCR assay but not ELISA assay. So the authors should state the results rigorously.